# Mutational signatures of non-homologous and polymerase theta-mediated end-joining in embryonic stem cells

Joost Schimmel ⓘ, Hanneke Kool, Robin van Schendel & Marcel Tijsterman* ⓘ

## Abstract

Cells employ potentially mutagenic DNA repair mechanisms to avoid the detrimental effects of chromosome breaks on cell survival. While classical non-homologous end-joining (cNHEJ) is largely error-free, alternative end-joining pathways have been described that are intrinsically mutagenic. Which end-joining mechanisms operate in germ and embryonic cells and thus contribute to heritable mutations found in congenital diseases is, however, still largely elusive. Here, we determined the genetic requirements for the repair of CRISPR/Cas9-induced chromosomal breaks of different configurations, and establish the mutational consequences. We find that cNHEJ and polymerase theta-mediated end-joining (TMEJ) act both parallel and redundant in mouse embryonic stem cells and account for virtually all end-joining activity. Surprisingly, mutagenic repair by polymerase theta (Pol θ, encoded by the *Polq* gene) is most prevalent for blunt double-strand breaks (DSBs), while cNHEJ dictates mutagenic repair of DSBs with protruding ends, in which the cNHEJ polymerases lambda and mu play minor roles. We conclude that cNHEJ-dependent repair of DSBs with protruding ends can explain *de novo* formation of tandem duplications in mammalian genomes.

**Keywords** cNHEJ; CRISPR/Cas9; double-strand break repair; embryonic stem cells; polymerase theta-mediated end joining
**Subject Categories** DNA Replication, Repair & Recombination; Stem Cells
**The EMBO Journal (2017) 36: 3634–3649**

## Introduction

Physical disruptions of chromosomes by means of DNA double-strand breaks (DSBs) are extremely hazardous to cells. If left unrepaired, DSBs can result in mis-segregation and loss of chromosomal fragments or induce an apoptotic response leading to cell death, while mutagenic repair of DSBs is a major source of disease-causing chromosomal rearrangements (Bunting & Nussenzweig, 2013). On the other hand, DSBs also directly contribute to the diversity of the immune system and to genome diversification, and thus to the evolution of organisms. Two dominant pathways of DSB repair, i.e. homologous recombination (HR) and classical non-homologous end-joining (cNHEJ), have been intensely studied and well characterized. In HR, 5′ end resection of broken chromosomes results in 3′ single-stranded DNA (ssDNA), which forms a filament with the RAD51 recombinase to invade an undamaged homologous sequence to guide error-free repair. In contrast, in cNHEJ, the ends of a DSB are protected against end resection through binding of the Ku70-Ku80 heterodimer, ultimately resulting in the joining of the two broken ends, catalysed by the DNA ligase IV (Lig4)–XRCC4 complex. Although cNHEJ is more prone to make mistakes, it provides cells with a dynamic and versatile mechanism to protect the genome if error-free HR is unavailable (Symington & Gautier, 2011; Betermier *et al*, 2014).

More recently, it became evident that cells can employ a highly mutagenic third pathway to repair DSBs, termed alternative end-joining (Alt-EJ), as it does not depend on cNHEJ factors (Deriano & Roth, 2013; Sfeir & Symington, 2015; Ceccaldi *et al*, 2016). This first came to light when end-joining activity was observed in cNHEJ-deficient *Saccharomyces cerevisiae;* error-prone DNA repair via this pathway was characterized by excessive deletions with small stretches of homology at the repair junctions (Boulton & Jackson, 1996). These findings provided a genetic basis for earlier work by Roth and Wilson (1986) who demonstrated the influence of micro-homologous pairing in end-joining in monkey cells. Similar observations were made in XRCC4- and Ku80-deficient hamster cells and in translocation junctions recovered from cNHEJ-deficient mice (Kabotyanski *et al*, 1998; Corneo *et al*, 2007). Early work demonstrated the involvement of XRCC1/DNA ligase III and PARP1 in Alt-EJ; more recently, the A-family DNA polymerase theta (Pol θ, encoded by the *Polq* gene) was identified as a quintessential component of Alt-EJ (Wang *et al*, 2006; Chan *et al*, 2010; Koole *et al*, 2014; Yousefzadeh *et al*, 2014; Ceccaldi *et al*, 2015; Mateos-Gomez *et al*, 2015). For reasons of clarity, we termed this repair mode TMEJ for polymerase theta-mediated end-joining (van Kregten & Tijsterman, 2014; Roerink *et al*, 2014) to set it apart from the umbrella term Alt-EJ, which can also encompass polymerase theta-independent microhomology-mediated end-joining. TMEJ of DSBs is

---

Department of Human Genetics, Leiden University Medical Center, Leiden, The Netherlands
*Corresponding author. Tel: +31 715269669; E-mail: m.tijsterman@lumc.nl

 

typified by the frequent presence of microhomology at the repair junctions, as well as the occasional manifestation of insertions of short stretches of DNA that originate from the flank of the DSB (Chan *et al*, 2010; Koole *et al*, 2014).

Pol θ and its related mutational signature are evolutionary conserved and were first described in *Drosophila melanogaster,* where Pol θ can repair DSBs induced by endonucleases or *P* element transposition (Chan *et al*, 2010). In the nematode *Caenorhabditis elegans*, Pol θ is required for the repair of replication stress-associated DSBs that form at persistent replication-blocking lesions and G-quadruplex structures in DNA (Koole *et al*, 2014; Roerink *et al*, 2014; Lemmens *et al*, 2015; van Schendel *et al*, 2015, 2016). Several lines of evidence have also implicated Pol θ in Alt-EJ in mouse cells: TMEJ provides resistance to DNA strand breaking agents in bone marrow stromal cells (Yousefzadeh *et al*, 2014), TMEJ-like footprints were observed at junctions of fused dysfunctional telomeres in embryonic fibroblasts (Mateos-Gomez *et al*, 2015), and TMEJ was found to regulate the repair of exogenous DNA substrates (Yousefzadeh *et al*, 2014; Wyatt *et al*, 2016). Mice deficient for Pol θ are viable but display hallmarks of increased genomic instability (Soulier *et al*, 2005). The fact that Pol θ uses microhomology in 3′ ssDNA protrusions to guide repair and predominantly acts on replication-associated breaks suggests that TMEJ can serve as an escape route for cells when error-free repair via HR is compromised. Indeed, human cancer cells deficient in HR were shown to rely on Pol θ functionality for their survival (Mateos-Gomez *et al*, 2015; Ceccaldi *et al*, 2016). In addition, Pol θ protein expression is often elevated in breast-cancer cells and this correlates with a poor prognosis for patients (Higgins *et al*, 2010; Lemee *et al*, 2010). Targeting Pol θ in patients carrying mutations in essential HR genes thus holds great promise in the clinic, by selectively killing of tumour cells.

Remarkably, while studies in zebrafish and worms point to a prominent role for TMEJ in DSB repair in germ cells and during embryogenesis (van Schendel *et al*, 2015; Thyme & Schier, 2016), this pathway is postulated to be a back-up mechanism in mammalian cells as it most clearly (if not exclusively) manifests under conditions where either HR or cNHEJ is compromised (Sfeir & Symington, 2015). Although this discrepancy may be explained by divergences in the regulation of DSB repair during evolution, we consider it more plausible that it may result from differences in the cellular context that is being studied: it has previously been shown that embryonic stem cells and somatic cells differently rely on distinct DSB repair mechanisms to maintain genomic integrity (Nagaria *et al*, 2013). In this light, it is noteworthy that mutational signatures found in genomic rearrangements that underlie congenital diseases are often characterized by microhomology, an indication for a causal role of Alt-EJ during gametogenesis or embryogenesis (Bunting & Nussenzweig, 2013).

Here, we have used non-transformed mouse embryonic stem cells deficient either in TMEJ, in cNHEJ or in both pathways, to study to what extent these different pathways of EJ contribute to DSB repair in mammalian embryonic stem cells and to the formation of DNA rearrangements. We used the CRISPR/Cas9 technology to introduce a single DSB in the selectable endogenous *HPRT* locus that is either blunt, or has ssDNA protrusions of different polarity. We determined the substrate specificities of cNHEJ and TMEJ, and elucidated how the configuration of the DSB dictates the nature of the resulting repair. In line with TMEJ signatures found in human

pathologies, we find that in embryonic stem cells TMEJ plays a prominent role also when HR and cNHEJ are functional. In addition and unexpectedly, we find that tandem duplications, important drivers of genome diversification and several human diseases (Thomas, 2005), can be explained by cNHEJ-mediated error-prone repair of DSBs with 3′ ssDNA protrusions.

# Results

## TMEJ and cNHEJ act redundant and in parallel in mouse embryonic stem cells

To study the contribution of both TMEJ and the cNHEJ pathway to the repair of DSBs in mammalian embryonic stem (ES) cells, we used CRISPR/Cas9 to make knockouts for *Polq* (TMEJ), *Ku80* and *Ligase4* (cNHEJ) in the 129/Ola-derived male E14 ES cell line (Robanus-Maandag *et al*, 1998). Using guide RNAs targeting distinct coding regions, we generated two different knockout cell lines per gene. Cell lines deficient for both end-joining pathways were generated by targeting the *Polq* gene in *Ku80*-knockout cell lines. Loss of Ku80 and Lig4 expression was confirmed by immunoblotting (Fig EV1A) (Zelensky *et al*, 2017), while loss of Pol θ in *Polq*-knockout cell lines was previously validated by complementation experiments with *Polq* cDNA (Zelensky *et al*, 2017). All results described in this study are obtained using two validated knockout cell lines per genotype. Although single mutant cell lines had growth characteristics similar to wild-type cells, double-knockout cell lines displayed reduced proliferation rates: cell-cycle distribution plots reveal more cells in G2/M (Fig EV1B). Cells that lack TMEJ or cNHEJ proved to be more sensitive towards ionizing radiation (IR) than wild-type cells, arguing that both pathways have independent functions to protect cells against DSBs (Fig 1A); cNHEJ appears to be the dominant pathway as Ku80- and Lig4-deficient cells are more profoundly affected by IR than Pol θ-deficient cells (Fig 1A). Cells that lack both end-joining pathways were found to be hypersensitive to IR (Fig 1A), which argues that these end-joining pathways also act redundantly. Mouse embryonic stem cells thus employ both end-joining pathways to respond to toxic DNA breaks. Moreover, in the absence of functional cNHEJ, these cells completely rely on Pol θ-mediated repair of DSBs to ensure survival but also vice versa.

## A selection-based DSB repair assay in mouse embryonic stem cells

IR causes DSBs by inducing clustered lesions on opposing DNA strands in close proximity of each other. The physical properties of IR result in the formation of a variety of DSBs, ranging from a blunt or near-blunt composition to DSBs with 5′ or 3′ protruding ssDNA segments of different lengths (Sage & Harrison, 2011). The degree and polarity of the protrusion potentially dictates end-joining pathway choice: while blunt DSBs can be easily repaired through direct ligation, DSBs with overhangs might induce or require enzymatic processing (fill-in or resection), or annealing of ssDNA at complementary sequences to stimulate repair. To study the contribution of EJ pathways to the repair of DSBs of different configurations, we have developed a selection-based assay that captures error-prone end-joining of a single genomic DSB. We used CRISPR/Cas9 to

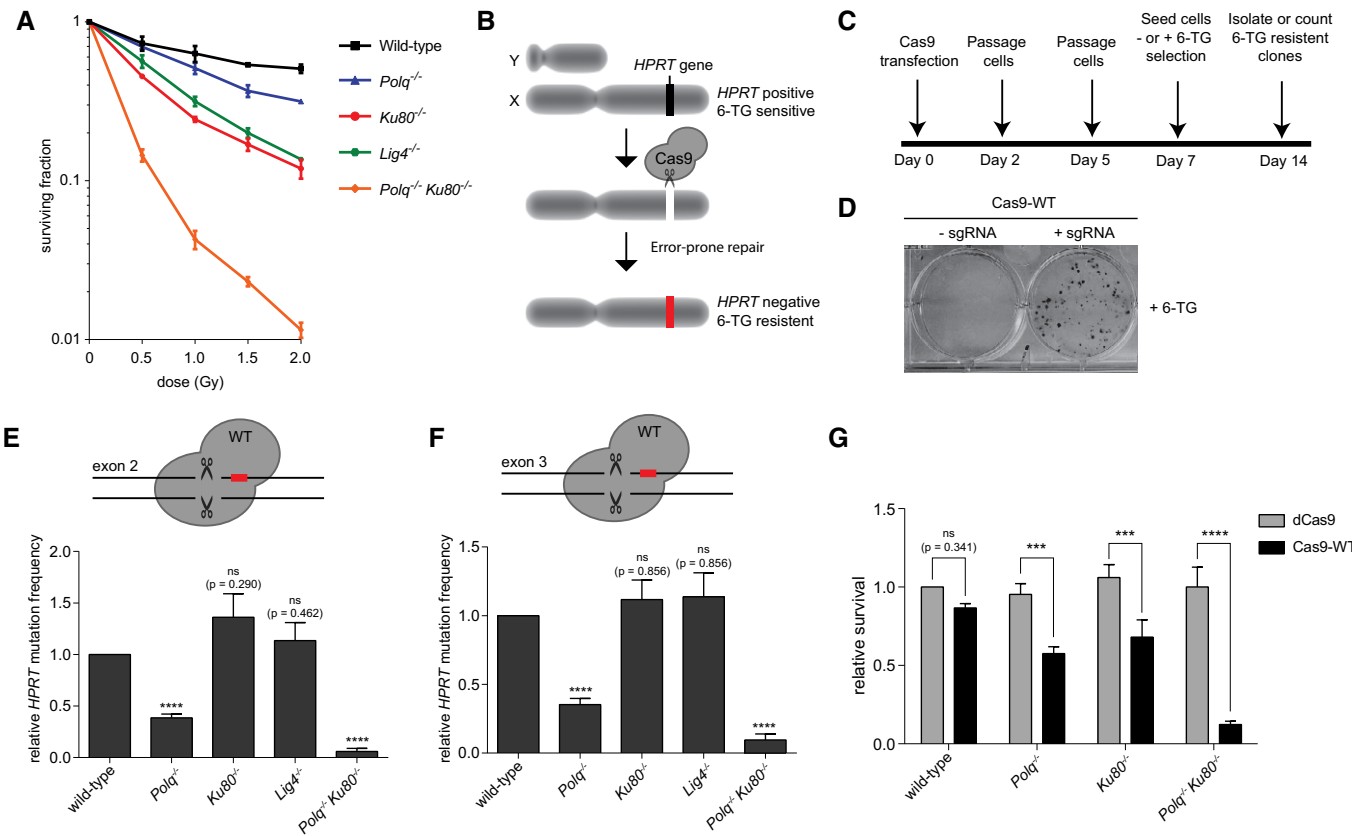

**Figure 1.  Embryonic stem cells use both TMEJ and cNHEJ to repair DSBs.**

A    Clonogenic survival of cell lines of the indicated genotypes after exposure to different doses of ionizing radiation (IR). Data shown are the mean $\pm$ SEM (wild-type, $Ku80^{-/-}$, $Lig4^{-/-}$, $Polq^{-/-}$ $Ku80^{-/-}$ cells $n = 4$, $Polq^{-/-}$ cells $n = 3$).

B, C   Schematic outline and timetable of the *HPRT* assay.

D    Methylene blue-stained dishes of cells that were transfected with wild-type Cas9 (Cas9-WT) only or Cas9-WT together with an *HPRT* sgRNA, subsequently cultured in 6-thioguanine (6-TG)-containing selection medium.

E, F   Relative *HPRT* mutation frequency for the indicated cell lines transfected with Cas9-WT targeting exon 2 (E) or Cas9-WT targeting exon 3 (F). The data shown represent the mean $\pm$ SEM ($n = 6$) and are expressed as a fraction of the mutation frequency observed in wild-type cells (set to 1). Statistical significance was calculated via unpaired *t*-test with Holm–Bonferroni correction. ns, not significant, ****$P < 0.0001$.

G    Relative cellular survival of the indicated cell lines transfected with Cas9-WT compared to the survival of cells transfected with nuclease-dead Cas9 (dCas9). The data shown represent the mean $\pm$ SEM ($n = 3$) and are expressed as a fraction of the survival observed in wild-type cells transfected with dCas9 (set to 1). Statistical significance was calculated by two-way ANOVA with Bonferroni correction. ns, not significant, ***$P < 0.001$, ****$P < 0.0001$.

induce a site-specific DSB in the selectable marker gene *Hypoxanthine-guanine phosphoribosyltransferase* (*HPRT*). HPRT is an enzyme involved in the synthesis of purines and can convert the drug 6-thioguanine (6-TG) to toxic thioguanine nucleotides, which induces cell death upon its incorporation into DNA. Loss of HPRT protein expression, as the result of mutagenic repair of a targeted DSB in the *HPRT* gene (induced by CRISPR/Cas9), would thus render cells resistant to 6-TG treatment (Fig 1B). This feature can be utilized to determine the *HPRT* mutation frequency, reflecting the efficiency of mutagenic repair of DSBs, and to analyse repair products (Fig 1C and D). Indeed, transfecting wild-type mouse ES cells with wild-type Cas9 (Cas9-WT) constructs co-expressing guide RNAs targeting either exon 2 or exon 3 of the *HPRT* gene (Fig EV1C) results in a robust induction of *HPRT* mutant cells; this is fully dependent on the enzymatic activity of Cas9 as expression of a catalytic dead Cas9 mutant (dCas9) did not result in a detectable *HPRT* mutation frequency (Fig EV1D and E).

## cNHEJ and TMEJ regulate double-strand break repair in embryonic stem cells

We next assayed the *HPRT* mutation frequency upon induction of predominantly blunt DSBs by Cas9-WT (Geisinger *et al*, 2016) in *Polq*, *Ku80*, *Lig4* and *Polq-Ku80* knockout cell lines and compared it to that in wild-type cells. We observed a strong reduction in the mutation frequency in *Polq* knockout cells as compared to wild-type cells for DSBs induced both in exon 2 and in exon 3 (2.6-fold and 2.8-fold reduction, respectively; Fig 1E and F). Depletion of Ku80 or Lig4 did not result in a significant change in the mutation frequency, suggesting that either cNHEJ is not contributing to error-prone repair or, alternatively, that TMEJ can completely compensate for the loss of cNHEJ. In support for the latter, we indeed found that mutation induction in $Ku80^{-/-}$ cells is almost entirely dependent on functional Pol θ (Fig 1E and F, $Polq^{-/-}$ $Ku80^{-/-}$ versus $Ku80^{-/-}$ cells). The observation that the mutation frequency in $Polq^{-/-}$

$Ku80^{-/-}$ cells is profoundly lower than that of $Polq^{-/-}$ single mutant cells argues for mutagenic cNHEJ in $Polq^{-/-}$, and potentially also in wild-type cells. Cas9-WT protein expression was comparable in all genotypes (Fig EV1F and G).

A greatly reduced number of *HPRT* mutant clones in $Polq^{-/-}$ $Ku80^{-/-}$ cells can be explained by envisioning error-free DSB repair acting as a back-up or, perhaps more likely, by cell death resulting from the inability to repair toxic and unresolved DSBs. To discriminate between these possibilities, we transfected wild-type, $Polq^{-/-}$, $Ku80^{-/-}$, $Lig4^{-/-}$, and $Polq^{-/-}$ $Ku80^{-/-}$ cells with GFP-tagged Cas9 versions and an sgRNA targeting *HPRT*. We then compared the clonogenic survival of GFP-positive cells transfected with Cas9-WT to GFP-positive cells transfected with dCas9. We found that DSB induction results in a mild decrease in clonogenic survival in $Polq^{-/-}$, $Ku80^{-/-}$, and $Lig4^{-/-}$ cells as compared to wild-type cells; however, double-deficient cells displayed severely reduced cellular survival (Figs 1G and EV1H). This result demonstrates that DSB induction directly interferes with proliferation in cells that lack both cNHEJ and TMEJ.

From these data, we conclude that in mouse ES cells: (i) virtually all mutagenic repair of CRISPR/Cas9-induced blunt DSBs is the combined result of TMEJ and cNHEJ, and (ii) that TMEJ can completely compensate for the loss of cNHEJ, but not vice versa: cNHEJ only acts redundantly to ~50% of TMEJ substrates.

## TMEJ and cNHEJ mutational signatures at blunt DSBs

The conclusion that error-prone repair of blunt DSBs is the summation of TMEJ and cNHEJ means that repair events in Pol θ-deficient cells reflect cNHEJ, while repair events in Ku80- and Lig4-deficient cells represent TMEJ. The *HPRT*-CRISPR assay thus also allows us to specifically study and compare the mutational signatures of these two end-joining pathways and then infer which pathway acts predominantly in wild-type cells. To that end, we sequenced the targeted region (~500 bp) of individually isolated clones that were resistant to 6-TG selection upon Cas9-WT-induced DSBs in *HPRT* exon 2 or exon 3 and analysed the repair products. The majority of mutations originating from blunt DSBs can be categorized into three main groups: (i) simple deletions, (ii) deletions accompanied by the insertion of DNA (delins) and (iii) insertion of DNA without loss of original sequence (insertion). For blunt DSBs introduced in *HPRT* exon 2 and exon 3, we collected in total 632 valid sequences from the different genetic backgrounds, in all of them simple deletions represented the largest class (Fig 2A and B, and Table EV1). Apart from minor sequence context-dependent differences in the mutation spectra of exon 2 and exon 3, which will not be discussed here (see also van Overbeek *et al*, 2016), a number of pathway-specific features become apparent. These concern (i) the usage of microhomology, (ii) the deletion size and (iii) the presence and nature of insertions at repair junctions.

First, the requirement for microhomology usage in TMEJ is close to absolute: 93% of all deletions isolated from $Ku80^{-/-}$- and $Lig4^{-/-}$-deficient cells have at least 1 nucleotide of microhomology (Figs 2C, and EV2A and C). The cases without microhomology might in fact result from residual Pol θ-independent repair (6–10% based on *HPRT* mutation frequency in $Polq^{-/-}$ $Ku80^{-/-}$ cells) that does not display microhomology usage (Fig 2C). Furthermore, the presence of small segments of repeated sequence immediately

upstream and downstream of the DSB appears to direct the mutagenic outcome in TMEJ towards removing the intervening sequence: at the exon 3 target, a stretch of 6 bp of possible microhomology make up for 43–53% of all cases in cNHEJ-deficient cells. In sharp contrast, microhomology does not influence cNHEJ as the distribution of homology in $Polq^{-/-}$ cells does not statistically deviate from a probable distribution.

Second, the size distribution of TMEJ repair products appears to be bimodal: apart from small 1- to 20-bp deletions that dominate the spectra in all genetic backgrounds, a class of larger deletions (> 50 bp) is found at both *HPRT* target sites, in $Ku80^{-/-}$ and $Lig4^{-/-}$ cells, but not in $Polq^{-/-}$ or $Polq^{-/-}$ $Ku80^{-/-}$ cells (Fig 2D), which indicates that these depend on Pol θ for their formation. In agreement with a role for Pol θ, virtually all these deletions show microhomology or templated inserts at the repair junction (Fig EV2B). Notably, this class is also observed in wild-type cells providing further support to the conclusion that TMEJ also acts when cNHEJ is functional.

Third, we observe clear manifestations of Pol θ-dependent extension of DSB-ends in which either the other DSB-end (*in trans*) or the extended end itself (*in cis*) is used as a template (Fig 2E). Interestingly, the inserts we observe in mouse ES cells closely resemble the TMEJ products analysed in *C. elegans* (van Schendel *et al*, 2016) and *Arabidopsis thaliana* (van Kregten *et al*, 2016), but are less similar to the more scrambled inserts found at Pol θ-dependent telomere fusions (Mateos-Gomez *et al*, 2015), a discrepancy we will discuss later. These so-called templated inserts are only present in wild-type and cNHEJ-deficient cells, not in TMEJ-deficient cells (Fig 2E), arguing for a causal involvement of Pol θ and again underwriting the notion of TMEJ activity in cNHEJ-proficient cells. In Pol θ-deficient cells, most inserts are only 1 or 2 base pairs long. While these inserts could arise from cNHEJ-mediated processing of near-blunt DSBs (which can have 1 or few nucleotides ssDNA protrusions), we also observe these products in $Polq^{-/-}$ $Ku80^{-/-}$ cells.

## Genetic requirements of EJ repair of DSBs with ssDNA protrusions

The analysis of repair products together with the *HPRT* mutation frequency data strongly argues for a prominent role for TMEJ in the repair of DSBs in embryonic stem cells. However, as shown in Fig 1A, cNHEJ functionality protects better against the cytotoxic effects of ionizing radiation. We hypothesized that this discrepancy may result from radiation-induced DSB configurations that are not blunt but instead have protrusions, whose repair may have different genetic requirements. To address this, we used Cas9-nickase constructs (Ran *et al*, 2013; Shen *et al*, 2014) to generate DSBs that have either a 46 nucleotide 5′ protrusion (Cas9-D10A) or a 46 nucleotide 3′ protrusion (Cas9-N863A) (Nishimasu *et al*, 2014) by introducing two single-strand breaks on opposing strands but 46 nucleotides apart in exon 2 of *HPRT* (Fig EV3A, sgRNA A and B). *HPRT* mutant cells were observed when both sgRNAs were co-expressed together with either Cas9-D10A or Cas9-N863A; the introduction of DSBs with a 5′ protrusion gives an eightfold higher *HPRT* mutation frequency than those with a 3′ protrusion (Fig EV3B and C). In both situations, the formation of *HPRT* mutant cells results from error-prone repair of DSBs, and not of single ssDNA nicks:

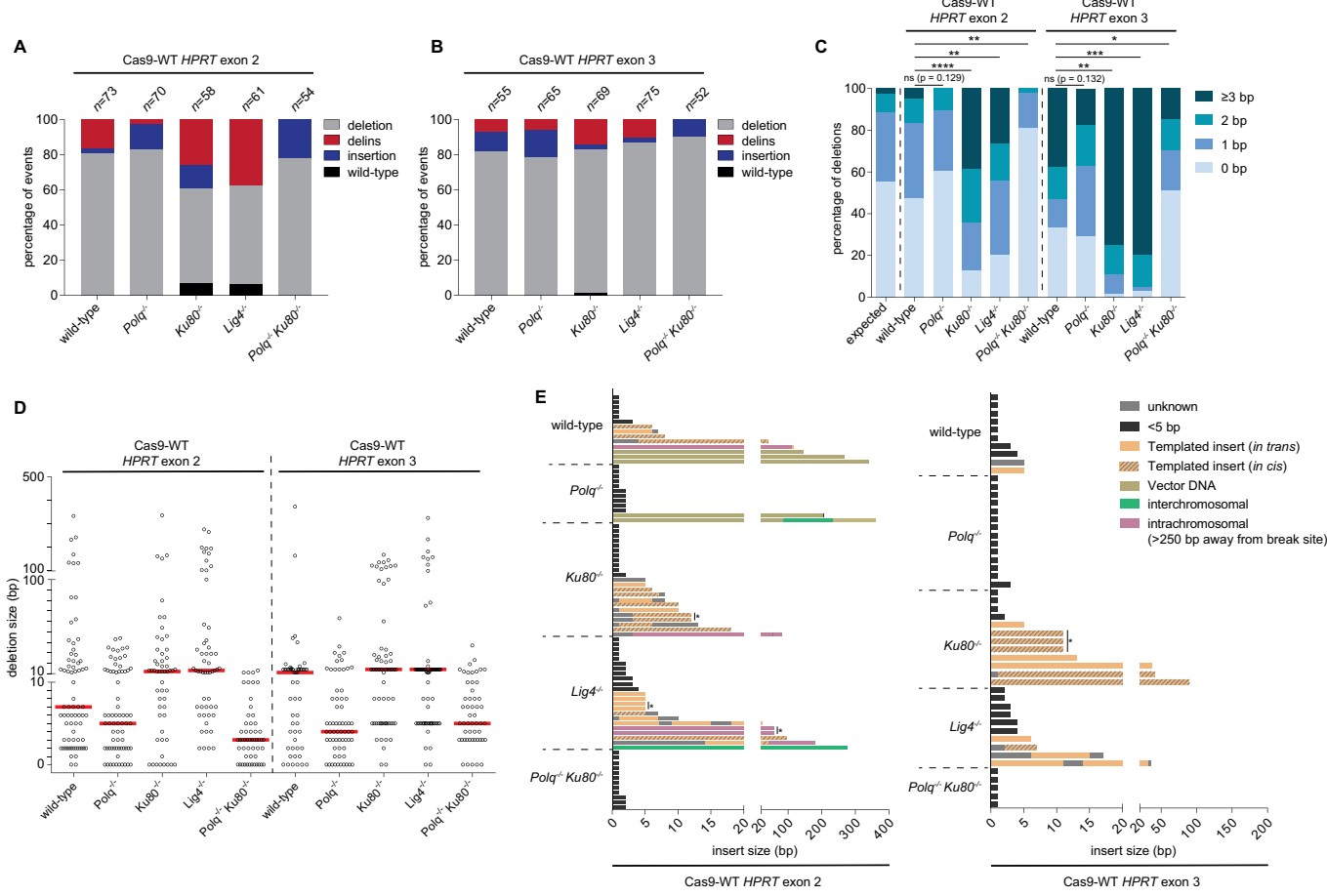

**Figure 2. Mutational signatures of cNHEJ and TMEJ on blunt DSBs.**

A, B    Column chart representation of *HPRT* repair products obtained from the indicated genomic backgrounds upon DSB induction with Cas9-WT in *HPRT* exon 2 (A) or exon 3 (B). The number of independently derived sequences per cell line is depicted above the columns. Events are classified in four distinct categories: (i) simple deletions (deletion), (ii) deletions accompanied by an inclusion of DNA (delins), (iii) insertion of DNA at the break site (insertion) and iv) sequences that did not contain a mutation in the amplified targeted region of *HPRT* (wild-type).

C    Quantification of the extent of microhomology for the category simple deletions for the indicated genotypes after induction of DSBs with Cas9-WT in *HPRT* exon 2 (left side of panel) or exon 3 (right side of panel). The first bar represents the distribution of microhomology that is expected for a randomly distributed set of deletions. Statistical significance was calculated via Mann–Whitney test with Holm–Bonferroni correction. ns, not significant, *$P < 0.05$, **$P < 0.01$, ***$P < 0.001$, ****$P < 0.0001$.

D    Plot of the deletion size of independently obtained *HPRT* repair events (described in A and B) from cell lines of the indicated genotype. Red lines indicate the median deletion size.

E    Graphical representation of the composition of DNA inserts for the categories delins and insertion obtained from the indicated cell lines for Cas9-WT-induced DSBs in *HPRT* exon 2 (left panel) or exon 3 (right panel). Inserts of DNA bigger than 4 bp were mapped and found to originate from either flanking sequences within 250 bp of the break site (templated inserts *in trans* and *in cis*), from sequences > 250 bp away from the break site (intrachromosomal), from sequences on different chromosomes (interchromosomal) or from sequences of the Cas9-WT px458 plasmid used for transfection (plasmid DNA). Some inserted sequences could not be mapped (unknown). Asterisks indicate identical inserts larger than 4 bp found within one cell line.

expression of only one of the two sgRNAs resulted in a barely detectable *HPRT* mutation frequency, likely because of efficient base-excision repair of single-strand nicks (Fig EV3B) (Dianov & Hubscher, 2013).

Next, we analysed and compared the *HPRT* mutation frequency resulting from DSBs with two categorically distinct protrusions in wild-type and end-joining deficient mouse ES cells. Equal expression levels of Cas9-variants were confirmed by immunoblotting (Fig EV3D and E). Figure 3 shows that inactivation of either TMEJ or cNHEJ barely affected the error-prone repair of DSBs with 5′ protrusions: the modestly reduced *HPRT* mutation frequency in

*Polq*$^{-/-}$ and Ku80$^{-/-}$ cells was statistically supported, the reduction in *Lig4*$^{-/-}$ cells was not (Fig 3A). However, the *HPRT* mutation frequency in *Polq*$^{-/-}$ Ku80$^{-/-}$ double-knockout cells (Fig 3A) drops two orders of magnitude. From this result, we conclude that (i) cNHEJ and TMEJ together are responsible for almost all error-prone repair of DBSs with sizable 5′ protrusions, and (ii) TMEJ and cNHEJ activity is almost completely redundant on these types of DSBs, which does not mean that the outcomes will be identical, as we will demonstrate later.

Also for these types of breaks, we determined the cellular survival of wild-type, *Polq*$^{-/-}$, *Ku80*$^{-/-}$, *Lig4*$^{-/-}$ and *Polq*$^{-/-}$

$Ku80^{-/-}$ cells upon DSB induction (Figs 3C and EV3F). While Ku80- and Pol θ-deficient cells are only mildly affected by DSB induction at the *HPRT* locus, we found that only 17% of cells that are deficient for both pathways survived to produce a colony (compared to $Polq^{-/-}$ $Ku80^{-/-}$ cells expressing dCas9; Fig 3C). This result, which is in line with the hypersensitivity of double-deficient cells to IR, suggests that the inability to repair a single DSB through either cNHEJ or TMEJ prohibits proliferation.

For the repair of DSBs with 3′ protrusions, we found a different involvement of TMEJ and cNHEJ, being that mutagenic repair is more affected by loss of Ku80 than by loss of Pol θ (Fig 3B). Mutagenic repair is virtually absent when both TMEJ and cNHEJ are disabled. While these data are in line with the greater demand on cNHEJ in cellular resistance against IR, it was unexpected considering current models about the substrate specificities of both pathways (Sfeir & Symington, 2015): the Ku complex preferentially binds blunt ends over ssDNA (Ristic *et al*, 2003; Foster *et al*, 2011), while purified Pol θ acts to extend minimally paired 3′ overhangs of duplexed DNA (Kent *et al*, 2015, 2016; Zahn *et al*, 2015). Also, genetic data hint towards TMEJ acting on 3′ protrusions DNA-ends (Wyatt *et al*, 2016). For possible explanations to reconcile our data with the existing literature, see the Discussion section.

Surprisingly, knocking out either cNHEJ, TMEJ or both pathways simultaneously had no effect on the cellular survival after induction of DSBs with 3′ protruding ends by Cas9-N863A (Figs 3C and EV3F). This, together with the relative low *HPRT* mutation frequency found after Cas9-N863A expression (Fig EV3B), suggests that either Cas9-N863A is less efficient than Cas9-D10A in nicking DNA or that DSBs with 3′ protruding ends are more easily shunted into a repair mode such as SSA that in this experiment setting does not result in a mutagenic outcome (as precise reannealing of the complementary protruding ends re-establishes the original DNA sequence).

## Mutational profiles of repair of DSBs with protruding ssDNA

While the experiments described above demonstrate that in mouse ES cells both TMEJ and cNHEJ can repair DSBs with protruding ends, they do not establish hierarchy: which pathway dominates? To address this question, we first determined the mutational signatures of each pathway at these types of breaks and then ask which signature is more abundant in wild-type cells. Because of the extremely low *HPRT* mutation frequency upon Cas9-D10A or Cas9-N863A expression in TMEJ and cNHEJ double-knockout cells (Fig 2B and C), we did not include these in the analysis.

For DSBs with 5′ ssDNA protruding ends, we found simple deletions with loss of bases on one or both ends to represent the biggest class of repair products in all genetic background examined (Figs 4A and EV4B, and Table EV1). The most noticeable differences between TMEJ and cNHEJ are as follows: (i) almost all deletions associated with TMEJ display microhomology at the repair junctions (Figs 4B, and EV4A and B), further augmenting the importance of terminal homology in Pol θ action; (ii) also for breaks that have 5′ protrusions TMEJ generates templated inserts *in cis* (Fig 4D); however, their signatures are very complex (e.g. inverse inserts while flanking templates were deleted) and too few cases were isolated to deduce clear properties or to infer mechanism. Most cases were observed in cNHEJ mutant cells, arguing again that Ku protect ends from rogue Pol θ activity; (iii) NHEJ products (in $Polq^{-/-}$ cells) have limited loss of 5′ protruding nucleotides (on average 10 bp on either side) and the vast majority of junctions are located within the sequence that produce the 5′ protruding end. TMEJ products displayed more substantial loss (35–40 bp on average; see Fig 4C: $Ku80^{-/-}$ and $Lig4^{-/-}$ cells) and many products had at least one junction outside the protruding sequence, likely because it is the 3′ hydroxyl end of the nick that can serve as a primer for Pol θ-mediated extension using the other DSB-end as a template. Because of the apparent

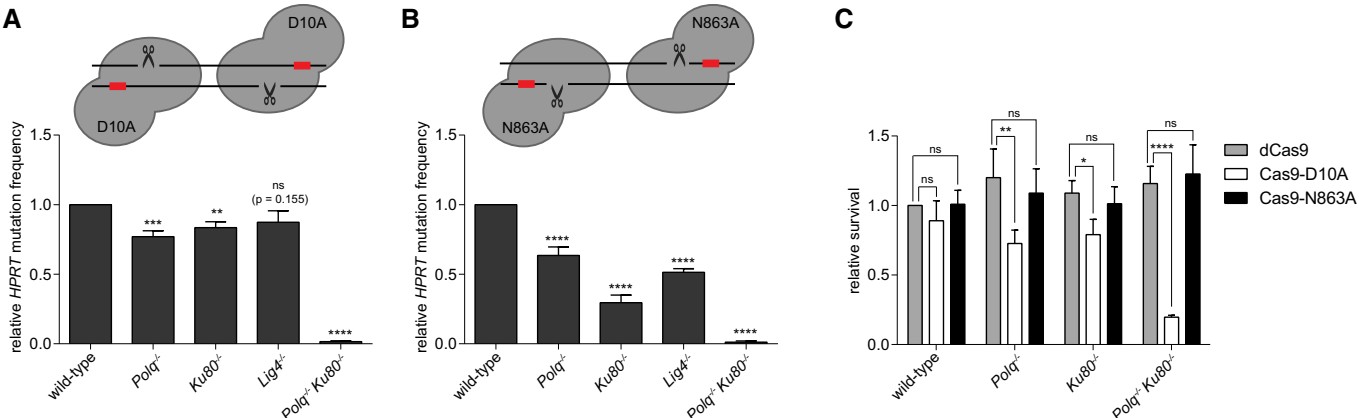

**Figure 3.  End-joining of DSBs with 5′ or 3′ ssDNA protrusions.**

A, B  Relative *HPRT* mutation frequency for the indicated cell lines transfected with Cas9-D10A (A) or Cas9-N863A (B), targeting exon 2 of *HPRT*. The data shown represent the mean ± SEM ($n \geq 4$) and are expressed as a fraction of the mutation frequency observed in wild-type cells (set to 1). Statistical significance was calculated via unpaired *t*-test with Holm–Bonferroni correction. ns, not significant, **$P < 0.01$, ***$P < 0.001$ ****$P < 0.0001$.

C  Relative cellular survival of the indicated cell lines transfected with the Cas9-nickase constructs compared to the survival of cells transfected with nuclease-dead Cas9 (dCas9). The data shown represent the mean ± SEM ($n = 3$) and are expressed as a fraction of the survival observed in wild-type cells transfected with dCas9 (set to 1). Statistical significance was calculated by two-way ANOVA with Bonferroni correction. ns, not significant, *$P < 0.05$, **$P < 0.01$, ****$P < 0.0001$.

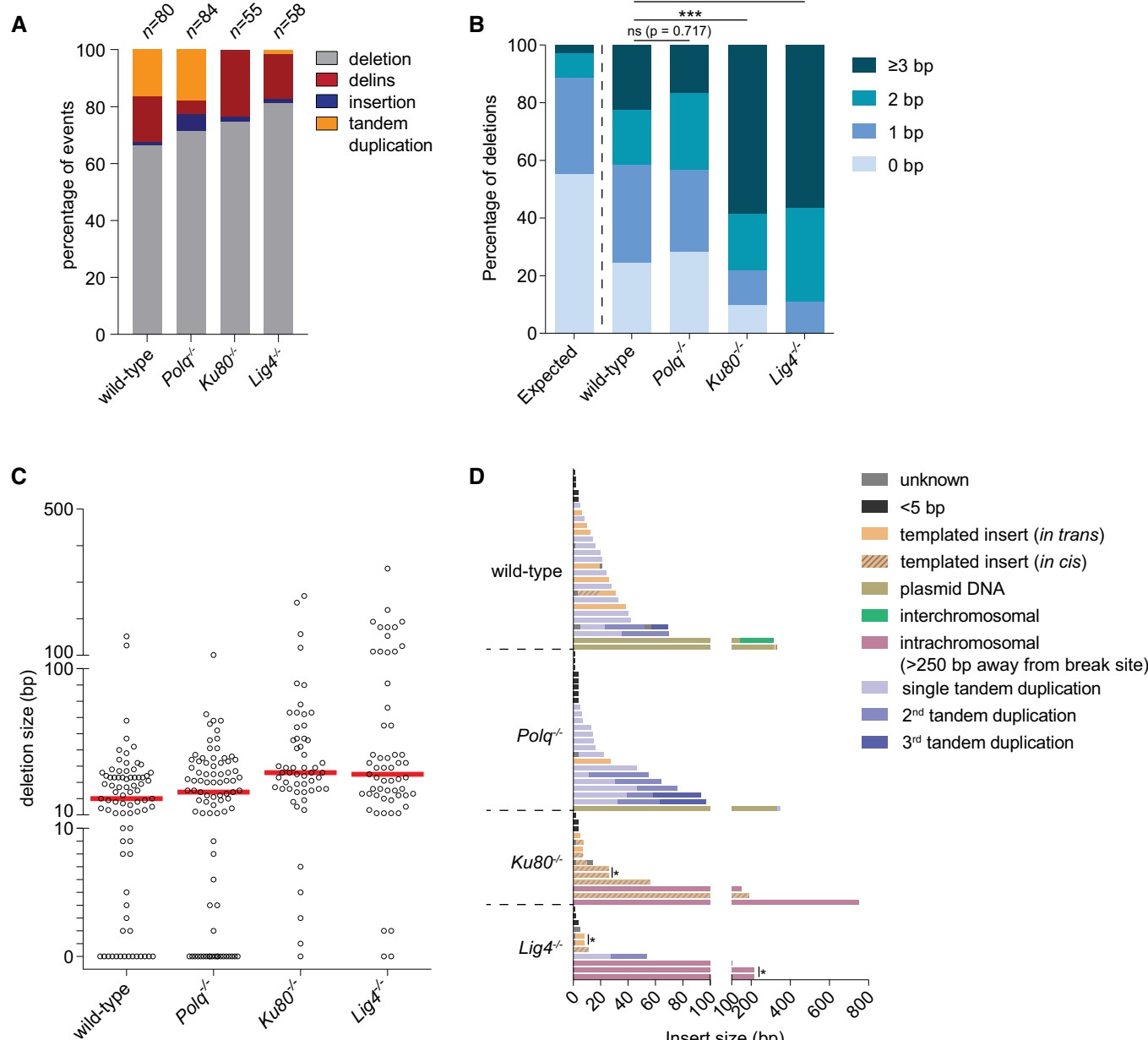

**Figure 4. Mutational signatures of cNHEJ and TMEJ on DSBs with a 5′ protrusion.**

A   Column chart representation of *HPRT* repair products obtained from the indicated genomic backgrounds after DSB induction with Cas9-D10A. In addition to the categories described in Fig 2A and B, the category "tandem duplication" is represented (see main text for details). The number of independently derived sequences is depicted above the columns.

B   Quantification of the degree of microhomology per genotype for the category simple deletions after induction of DSBs with Cas9-D10A. Statistical significance compared to the distribution in wild-type cells was calculated via Mann–Whitney test with Holm–Bonferroni correction. ns, not significant, ***$P < 0.001$, ****$P < 0.0001$.

C   Plot of the deletion size per indicated genotype of independently obtained *HPRT* repair events (described in A). Red lines indicate the median deletion size.

D   Graphical representation of the composition of DNA inserts for the categories delins, insertion and tandem duplications depicted in (A). The formation of tandem duplications can result in multiple rounds of cutting and repair, which we represented by 2nd and 3rd tandem duplications. Asterisks indicate identical inserts (except for single tandem duplications) larger than 4 bp found within one cell line.

stability of the complementary protruding ends, a category of mutations manifest that were not found in the spectra of blunt DSBs, i.e. tandem duplications (TDs), which we here define as direct repeats of sequences that originate from the protruding segment. A TD can

result from the repair of DSBs with protruding tails if some of the protruding ssDNA at both ends of the break is converted to dsDNA during the repair reaction (prior to or after joining). For instance, if at both ends of a DSB with 5′ protruding tails a DNA polymerase

initiates DNA synthesis at the dsDNA–ssDNA junction using the complete ssDNA protrusions as a template, then subsequent joining will result in a TD comprising the sequence that is in between the nicks. The length of such a TD can be countered by the activity of 5′ to 3′ exonucleases (see Appendix Fig S1A for a possible model). Although TDs resemble Pol θ-mediated templated inserts, they are formed by cNHEJ and not via TMEJ: while TDs are equally frequent in *Polq*$^{-/-}$ and wild-type cells (17.9 and 16.3% of the cases, respectively), they are nearly absent in mutation spectra derived from *Ku80*$^{-/-}$ and *Lig4*$^{-/-}$ cells (Fig 4A). Off note, because *in vivo* TD formation does not eliminate the sgRNA targets, consecutive rounds of repair and nicking can result in complex footprints with multiple repeated sequences (Fig 4D), which hampers unambiguous annotation of a few signatures. We suspect that the single case in *Polq*$^{-/-}$ cells that is annotated as templated insert to be the result of a deletion formed after re-cutting of a TD outcome (Fig 4D and Appendix Fig S1B).

By comparing the spectrum of mutations derived from wild-type cells to the mutational signatures of TMEJ and cNHEJ, we conclude that the vast majority of DSBs with 5′ ssDNA protruding ends are repaired through cNHEJ in genetically uncompromised mouse ES cells: on basis of rearrangement type, homology usage, deletion size and presence or absence of inserts, the wild-type spectrum is very similar to the spectrum obtained from *Polq*$^{-/-}$ cells, yet very different to the spectra of Ku80- and Lig4-deficient cells. Perhaps remarkably, a similar conclusion can be drawn for DSBs with 3′ ssDNA protruding ends (Fig 5 and Table EV1): the fast majority of repair events derived from wild-type and *Polq*$^{-/-}$ cells represent TDs (80.9 and 82.9% of the cases, respectively), yet in *Ku80*$^{-/-}$ cells, we observe a twofold reduction as compared to wild-type cells; depletion of Lig4 also reduces the number of TDs (Fig 5A). For TDs generated at DSBs with 3′ protrusions, we envisage different biochemistry than for TDs at DSBs with 5′ protrusions, i.e. extension of the 3′ protrusion at one end of the break using the 3′ protrusion at the other end as a template generates a TD that is defined by the position where both protrusions align (see also the model in Fig 6). Compared to TDs generated by cNHEJ, TDs generated by TMEJ have different characteristics: (i) single tandem duplications in Ku80- and Lig4-deficient cells are almost all associated with microhomology (Figs 5B and EV5); (ii) are in general larger, thus a larger proportion of the 3′ protruding segment is retained (Fig 5C); and (iii) are occasionally accompanied by *in cis* templated inserts (Fig 5D). The dominant presence of TDs in wild-type cells that result from cNHEJ activity on DSBs with ssDNA protruding ends may shed new light on the aetiology of small TDs that are abundantly found in evolving genomes (Messer & Arndt, 2007); their configuration is identical to the product of cNHEJ we here describe.

The formation of TDs at break sites can be explained by *de novo* DNA synthesis in which DNA polymerases use both protruding ends as a template. While Pol θ is likely to provide this enzymatic activity in TMEJ of DSBs with 3′ protruding ends (Kent *et al*, 2015, 2016), the protein(s) to serve this function in cNHEJ has not been identified. Possible candidates are polymerase lambda (Pol λ, encoded by the *Poll* gene) and polymerase mu (Pol μ, encoded by the *Polm* gene), two DNA polymerases previously demonstrated to act in cNHEJ (Ramsden & Asagoshi, 2012; Waters *et al*, 2014). To test this idea, we created *Poll*$^{-/-}$ and

*Polm*$^{-/-}$ single- and *Poll*$^{-/-}$ *Polm*$^{-/-}$ double-mutant cells using CRISPR/Cas9 technology and confirmed loss of Pol λ and Pol μ expression in independently obtained clones (Appendix Fig S2A). We next established that these polymerases also act in DSB repair in mouse ES cells by demonstrating an IR-hypersensitive phenotype for *Poll*$^{-/-}$ *Polm*$^{-/-}$ cells (Fig 6A). While *Polm*$^{-/-}$ single mutant cells display increased sensitivity to IR, knocking out *Poll*$^{-/-}$ only confers hypersensitivity in a *Polm*$^{-/-}$ genetic background. *Poll*$^{-/-}$ *Polm*$^{-/-}$ cells are, however, less sensitive to IR than *Ku80*$^{-/-}$ cells. All these observations are in agreement with previous work and provide support for considering these cell lines to behave as functional nulls (Vermeulen *et al*, 2007; Lucas *et al*, 2009; Pryor *et al*, 2015).

Next, we used these cells to address the question whether Pol μ and Pol λ act in cNHEJ-mediated TD formation by analysing the mutation frequencies and spectra after introducing DSBs with protruding ends (Fig EV3A). Cells lacking one or both polymerases had an *HPRT* mutation frequency comparable to wild-type for DSBs with either 5′ ssDNA or 3′ ssDNA protruding ends (Fig 6B and E, respectively) arguing that mutagenic repair does not depend on Pol μ and/or Pol λ. However, for a small number of features the mutation signature deviates from Pol μ- and Pol λ-proficient cells pointing towards a modest yet interesting role: for DSBs with 5′ protrusions, we found that TDs are formed less frequently (6.9% of the cases versus 16.25% in wild-type cells) (Fig 6C), and deletions are more frequently associated with microhomology (Fig 6D). Although the frequency of TD formation did not change for DSBs with 3′ protrusions (Fig 6F), we here also found a reduction in the amount of TDs without homology in both *Polm*$^{-/-}$ and *Poll*$^{-/-}$ *Polm*$^{-/-}$ cells, as compared to wild-type cells (18.75 and 14.29% versus 40%, respectively) (Fig 6G). The observed increased manifestation of microhomology is the result of cNHEJ (and not TMEJ) action, as we also found it in a Pol θ mutant background (Table EV1). While our data thus point to a modest yet detectable role for Pol λ and Pol μ in the repair of CRISPR/Cas9-induced DSBs with protruding ends, it also reveals (by deduction) that an activity must exist within cNHEJ that helps to repair such DSBs using microhomology (Fig 6H).

## Discussion

Here, by introducing a single endogenous chromosomal break in the *HPRT* gene, we studied the contribution of two distinct end-joining pathways to the repair of different types of DSBs and defined their mutational outcomes. We show that in embryonic stem cells TMEJ acts parallel as well as redundant to cNHEJ, enabling cellular resistance to genomic insults at the expense of mutations. Also in cNHEJ-proficient cells, mutational signatures are observed that can be ascribed to the action of Pol θ. Surprisingly, mutagenic TMEJ is most prevalent for DSBs that are blunt, while mutagenic repair of DSBs that have 5′ or 3′ protruding is predominantly regulated by cNHEJ. Detailed analysis of repair products revealed a yet unidentified cNHEJ-dependent repair mode for DSBs with 3′ protruding ends. We demonstrate that TMEJ and cNHEJ together constitute the error-prone mechanisms by which embryonic stem cells repair DSBs. Their contribution to

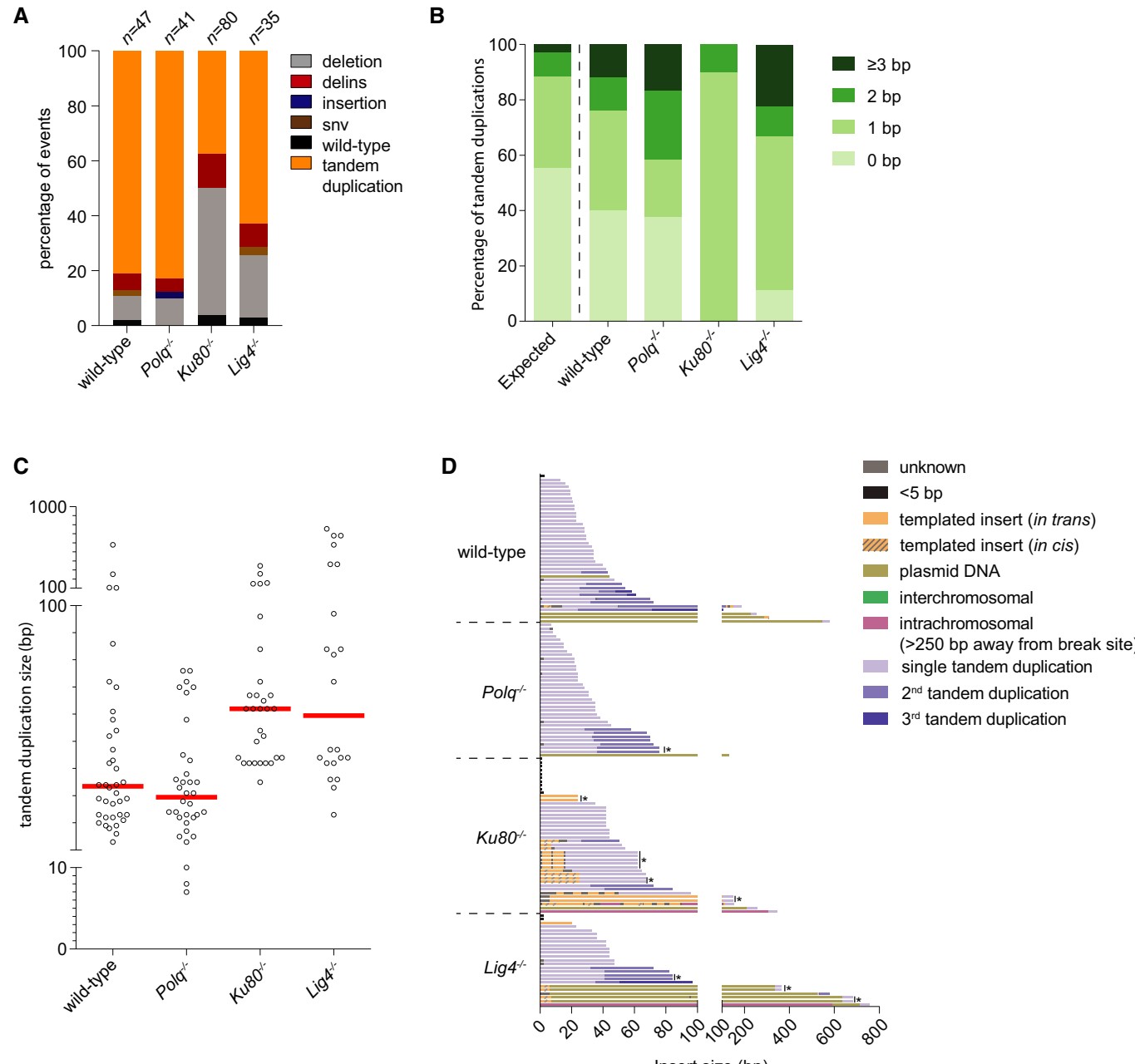

**Figure 5.  Mutational signatures of cNHEJ and TMEJ on DSBs with a 3′ protrusion.**

A  Column chart representation of *HPRT* repair products obtained from the indicated genotype upon repair of DSBs induced with Cas9-N863A. The number of independently derived sequences per cell line is depicted above the columns. SNV, single nucleotide variant.

B  Quantification of the degree of microhomology for the category tandem duplications induced by Cas9-N863A, but only for those that classify as single duplication events, genotypes of the cell lines are indicated.

C  Plot of the size of the duplicated segment for events in the category "tandem duplication" depicted in (A). Red lines indicate the median size.

D  Graphical representations of the composition of DNA inserts for the categories delins, insertion and tandem duplications depicted in (A). Asterisks indicate identical inserts (except for single tandem duplications) larger than 4 bp found within one cell line.

error-prone but also error-free repair can explain why cells become extremely sensitive to IR-induced DSBs when both pathways are corrupted. Even more striking, the induction of a single endogenous DSB through the introduction of two nicks on opposing strands leads to cell death in cells that are double deficient in TMEJ and cNHEJ.

## TMEJ acts parallel to cNHEJ in embryonic cells

Our genetic analysis supports the notion that TMEJ is responsible for the vast majority of DSB repair activity that was previously attributed to Alt-EJ. By exploring more phenomena associated with DSB repair [e.g. class switch recombination (Yousefzadeh *et al*,

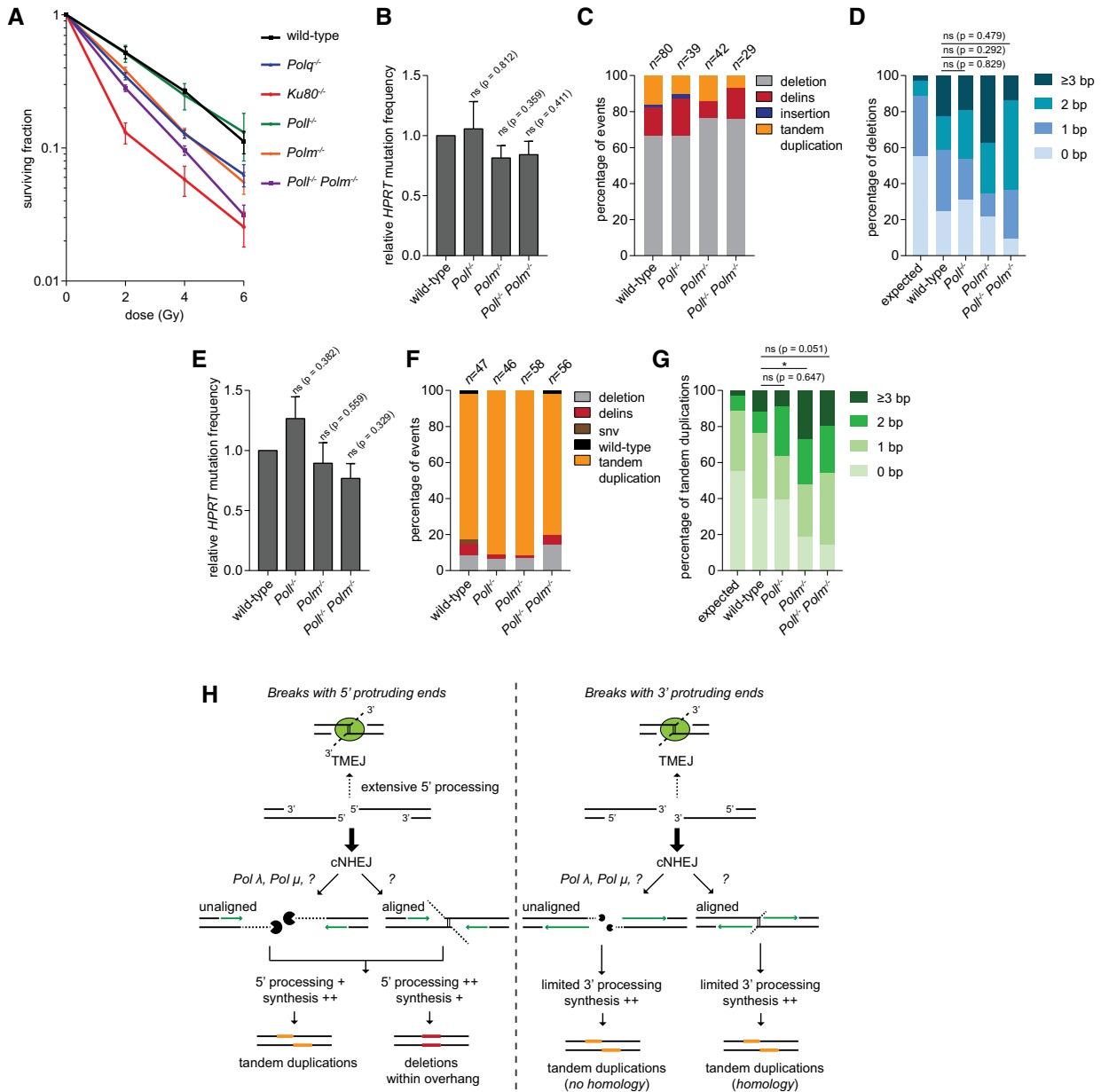

**Figure 6. Analysis of DSB repair in Pol λ- and Pol μ-deficient cells.**

A   Clonogenic survival of cell lines of the indicated genotypes after exposure to different doses of ionizing radiation (IR). Data shown are the mean ± SEM (n = 4).

B   Relative *HPRT* mutation frequency for the indicated cell lines transfected with Cas9-D10A, targeting exon 2 of *HPRT*. The data shown represent the mean ± SEM (n = 4) and are expressed as a fraction of the mutation frequency observed in wild-type cells (set to 1). Statistical significance was calculated via unpaired *t*-test with Holm–Bonferroni correction.

C   Column chart representation of *HPRT* repair products obtained from indicated genomic backgrounds after DSB induction with Cas9-D10A, compared to the previously obtained products from wild-type cells (Fig 4). The number of independently derived sequences per cell line is depicted above the columns.

D   Quantification of the degree of microhomology per genotype for the category simple deletions after induction of DSBs with Cas9-D10A. Statistical significance compared to the distribution in wild-type cells was calculated via Mann–Whitney test with Holm–Bonferroni correction.

E   Relative *HPRT* mutation frequency for the indicated cell lines transfected with Cas9-N863A, targeting exon 2 of *HPRT*. The data shown represent the mean ± SEM (n = 4) and are expressed as a fraction of the mutation frequency observed in wild-type cells (set to 1). Statistical significance was calculated via unpaired *t*-test with Holm–Bonferroni correction.

F   Column chart representation of obtained *HPRT* repair products per genotype upon repair of DSBs induced with Cas9-N863A, compared to the previously obtained products from wild-type cells (Fig 5). The number of independently derived sequences per cell line is depicted above the columns.

G   Quantification of the degree of microhomology for the category tandem duplications induced by Cas9-N863A, but only for those cases that classify as a single duplication event; genotypes of the cell lines are indicated. Statistical significance compared to the distribution in wild-type cells was calculated via Mann–Whitney test with Holm–Bonferroni correction. ns, not significant, *P < 0.05.

H   Tentative model for mutagenic repair of DSBs with protruding ends (see text for details).

2014)] and, here, by deconvoluting the substrate specificities of EJ pathways, it is becoming increasingly evident that TMEJ also acts as a first-line defence mechanism against DSBs in specific biological contexts. The clear contribution of TMEJ in genetically non-compromised embryonic stem cells agrees well with the observation that DSB-induced translocation formation in mouse embryonic cells and in mice models is mostly cNHEJ independent (Weinstock *et al*, 2007; Simsek & Jasin, 2010; Mateos-Gomez *et al*, 2015). Intriguingly, the apparent shift in end-joining pathway usage towards TMEJ in cells that have a more stem-cell like character is also found in other species: in the model systems *C. elegans* and zebrafish TMEJ is the dominant mode of (error-prone) DSB repair in germ and embryonic cells; somatic cells, however, employ mostly cNHEJ (van Schendel *et al*, 2015; Thyme & Schier, 2016). In agreement, Wyatt and colleagues recently showed that mouse embryonic fibroblasts (MEFs) preferentially use cNHEJ to repair Cas9-induced blunt DSBs leading them to conclude that TMEJ functions merely as a back-up mechanism. In addition, when comparing those data with ours, larger deletions were observed at break sites in cNHEJ-deficient fibroblast as compared to stem cells, which could argue that in cNHEJ-deficient somatic cells DNA-ends are susceptible to trimming before TMEJ can engage.

It thus appears that TMEJ is especially important in pluripotent fast-cycling cells, where robust repair mechanisms are needed to maintain the capacity of self-renewal (Nagaria *et al*, 2013) and cNHEJ activity is relatively low (Tichy *et al*, 2010). In contrast to somatic cells, embryonic stem cells are mostly in S-phase (Savatier *et al*, 2002), a stage of the cell cycle that is geared towards using HR to repair DSBs (Tichy *et al*, 2010). We speculate that particularly DSBs that occur in S-phase but cannot use a non-damaged sister chromatid as a repair donor, may heavily rely on TMEJ. These could be DSBs that occur in DNA segments that are not yet replicated, or DSBs induced at sites opposite a damaged sister chromatid (Lemmens *et al*, 2015). Also transposon mobilization, when co-occurring at both sister chromatids, may be a physiological source of DSBs that completely rely on TMEJ for their repair (van Schendel *et al*, 2015). In good agreement with this hypothesis, it was previously shown that especially replication-associated breaks are targets for TMEJ (Roerink *et al*, 2014), that microhomology-mediated repair requires similar end resection as HR (Truong *et al*, 2013) and that the inhibition of HR factors induced a mutational signature with prominent microhomology usage, which typifies Pol θ action (Ahrabi *et al*, 2016). The formation of 3′ protruding ssDNA tails by end resection during HR might thus be a prerequisite for TMEJ; it was indeed recently found that purified Pol θ can pair and extend the 3′ ssDNA tails of dsDNA duplexes (Kent *et al*, 2015). Interestingly, and in seemingly contrast to this rationale, we found that not TMEJ, but cNHEJ preferentially repairs endogenous DSBs that are generated to contain 3′ ssDNA protrusions. However, it may very well be that repair of 3′ overhangs that result from two distinct nicks in opposing strands (in, e.g., G1 phase cells) requires different enzymology than 3′ overhangs generated by end resection of a blunt DSB in S-phase (for which we observe a 60% dependency on TMEJ). This rational fits well with the recent finding that blunt DSBs result in higher levels of HDR-mediated repair compared to DSBs that have protruding ends (Vriend *et al*, 2016). Perhaps the size of the protrusion may also be a factor of relevance in DSB repair

pathway choice: it was recently found in MEFs that cNHEJ joins exogenous linear DNA substrates with 45-nt 3′ ssDNA overhangs, yet TMEJ is more important for substrates with overhangs of 70 nt (Wyatt *et al*, 2016). Although we observe a similar genetic dependency for Cas9-induced genomic DSBs with 46-nt 3′ ssDNA overhangs, there are also marked differences: we here found that cNHEJ repairs genomic breaks in a manner in which the DNA sequence in the overhangs is retained, while this was not the outcome in the aforementioned study. It may be that 3′ ssDNA overhangs created in the genome by Cas9 are held together and/or are well protected against exonuclease activity, and as such may be differently processed than linear DNA substrates transfected into cells. We should also note that our selection-based assays do not allow us to address the contribution of cNHEJ and TMEJ to error-free repair as this would restore the original non-mutant sequence, perhaps leading to cycles of repair and re-cutting as long as Cas9 is expressed.

### Mutational signatures of EJ pathways

Detailed analysis of repair products revealed that both end-joining pathways result in remarkably similar genomic scars, yet with a number of distinct features. In line with mutational analysis in different biological systems (reviewed in Black *et al*, 2016; van Kregten *et al*, 2016), we found that TMEJ (but not cNHEJ) in mouse embryonic stem cells is characterized by microhomology and by the occasional insertion of locally derived DNA. Three modes of Pol θ action have recently been described for purified mammalian Pol θ (Kent *et al*, 2016). In one of these modes, i.e. templated extension *in cis*, protruding ssDNA snaps back on itself to act as a template for repair. Manifestations of this mode are most prominent in cNHEJ-deficient cells. The binding of Ku to DNA-ends thus not only serves to inhibit HR (by limiting end resection) and to stimulate cNHEJ, but also acts to prevent ssDNA of folding back on itself. Similar to what has been demonstrated for RPA in yeast, Ku can thus act to suppress microhomology-mediated formation of hairpin-capped DNA-ends that can induce gross chromosomal rearrangements (Chen *et al*, 2013). Besides these *in cis* generated products, templated inserts resulting from so-called *in trans* reactions are present: futile cycles of TMEJ where one DSB-end served as the primer and the other as the template for Pol θ-mediated extension (van Schendel *et al*, 2016). However, little evidence was found for the proposed terminal transferase activity of Pol θ, which is demonstrated *in vitro* and is also suggested to explain the more scrambled inserts found at Pol θ-mediated telomere fusions (Mateos-Gomez *et al*, 2015). Perhaps, the very restricted possibility of pairing the TTAGGG repeat sequence of two telomeric 3′ overhangs is disfavouring efficient usage of a repair mode that is strongly stimulated by microhomology of at least three base pairs (Wyatt *et al*, 2016). This notion also provides a sequence-context explanation for the observation that telomeric fusions in repair proficient cells are grosso modo the result of cNHEJ action (Celli *et al*, 2006). In such a scenario, Pol θ's terminal transferase activity may be needed to fortuitously create a degree of microhomology sufficient for extension. The sequence context of a typical CRISPR-induced DSB, however, is such that minimal homology of the terminal bases is virtually always present in close proximity of the break ends.

## cNHEJ-mediated repair of DSBs with overhangs as a source of tandem duplications

We found that tandem duplications are a prominent outcome of DSB repair in embryonic stem cells, in line with recent observations made in *A. thaliana* and human somatic cells (Schiml *et al*, 2016; Bothmer *et al*, 2017). Tandem duplications are the most common form of small DNA insertions observed in the human genome (Messer & Arndt, 2007), thereby contributing to genome expansion and thus evolution but also resulting in novel gene functions in human diseases. FLT3-internal tandem duplications, for example, are often found in patients with acute myeloid leukaemia and their occurrence is linked to a poor prognosis (Levis & Small, 2003). Originally it was believed that tandem duplications form due to replication slippage or unequal crossing over (Levinson & Gutman, 1987); however, more recent data suggest that tandem duplications may be generated by erroneous repair of DSBs that result from two adjacent single-strand nicks on opposing strands (Messer & Arndt, 2007; Vaughn & Bennetzen, 2014; Schiml *et al*, 2016; Bothmer *et al*, 2017).

We propose that for DSBs with 5′ protrusions the mutational signature results from two opposite enzymatic activities: (i) *de novo* DNA synthesis at the 3′ terminus, which is templated by the 5′ protrusion, and (ii) resection of the 5′ ssDNA overhangs, thus restricting the size of the template. The summation of both activities at either DSB-end explains the two main classes of mutations that are observed: when resection at the 5′ end exceeds polymerase action on the 3′ end, deletions are produced that can still contain segments of the original overhang; however, when DNA synthesis at the 3′ end exceeds 5′ resection, tandem duplications of different sizes occur (Fig 6H). These intermediate molecules can *a priori* be repaired via blunt end ligation or via annealing between homologues sequences within the overhang followed by gap filling.

Intriguingly, mutagenic repair of DSBs with 3′ ssDNA overhangs mainly results in tandem duplications. Deletions may be less likely in these situations because 3′ ssDNA protrusions are less susceptible to resection perhaps aided by protective binding of proteins [e.g. RPA (Chen *et al*, 2013)]. A similar processing of DSBs with 3′ ssDNA protrusions was recently observed in human cancer cells (Bothmer *et al*, 2017). That study also demonstrated that 5′ ssDNA protrusions are resected to produce 3′ ssDNA protrusions which can then engage a homologous gene on the same chromosome to provoke a gene conversion event. In our study, we can also infer complete resection of 5′ ssDNA protruding ends (Fig EV4B) to liberate the 3′ hydroxyl ends required for TMEJ (Fig 6H). Interestingly, Bothmer *et al* perturbed the formation of Cas9-N863A-induced tandem duplications (and Cas9-D10A-induced gene conversion) by ectopic expression of the 3′ exonuclease TREX2. This observation provides experimental support for the notion that the stability of the 3′ ssDNA protrusions is a key determinant in the formation of tandem duplications. We here provide experimental evidence for a major role of cNHEJ in the formation of such tandem duplications. We conclude from the reduced *HPRT* mutation frequency in *ku80*$^{-/-}$ and *lig4*$^{-/-}$ cells as well as from the deviant mutation spectra in these cells that tandem duplications are formed via a cNHEJ-dependent mechanism. Which polymerase facilitates DNA synthesis remains an enigmatic question. Pol λ and Pol μ have been extensively studied using substrates with short overhangs: whereas Pol λ

can fill-in short gaps at terminally aligned breaks, Pol μ has the ability to add nucleotides to 3′ overhangs to potentially generate microhomology [(Ramsden & Asagoshi, 2012; Chang *et al*, 2017) and references therein]. While Pol λ and Pol μ facilitate repair of DSBs with small protrusions (Pryor *et al*, 2015), recent work demonstrated that depletion of these polymerases did not decrease end-joining of transfected DNA substrates with more extensive, aligned, 3′ ssDNA overhangs (Wyatt *et al*, 2016). In agreement, we found that depletion of polymerase lambda and mu in mES cells did not affect the efficiency of mutagenic repair of DSBs with ssDNA protrusions. However, we found evidence for a more subtle involvement of polymerase lambda and mu as the mutation profiles in cells lacking these two polymerases are different than those in wild-type cells. Our data are consistent with a notion of a greater demand for these cNHEJ polymerases at DSBs with unaligned overhangs (Fig 6H). More work is required in particular using DSBs at multiple genomic sites to investigate a potential sequence context-dependent involvement.

### Concluding remarks

As the genetic landscape of different cancer cells is being mapped, it is becoming increasingly important to link mutational signatures to specific DNA repair pathways (Helleday *et al*, 2014). In this work, we have assessed the contribution of two distinct end-joining pathways to the repair of chromosomal breaks as well as determined their individual contribution to specific mutation types. The pathway-specific signatures that we have revealed can be used in typifying cancer cells, for instance to identify tumours for which Pol θ-mediated repair became the "last resort" to repair toxic chromosome breaks (Ceccaldi *et al*, 2015; Mateos-Gomez *et al*, 2015). Profiling cancer cells based on mutational signatures can help in deciding treatment choice, especially in cases where the cells are suspected to have a defect in HR, for instance in individuals that carry a variant of unknown significance in HR genes (Guidugli *et al*, 2014). We also realize that the assays described here provide a highly specific platform to assay the potential specificity of many recently identified modifiers of end-joining repair (e.g. reviewed in Deriano & Roth, 2013) and to, through genetic screening, identify novel factors in either pathway. We foresee that an improved understanding of end-joining pathways to genome stability can help to identify potential pathway-specific strategies for (combinatorial) treatments of human cancers as was recently suggested for Pol θ (Killock, 2015; Beagan & McVey, 2016; Dai *et al*, 2016; Wood & Doublie, 2016).

# Materials and Methods

### ES cell culture

129/Ola-derived IB10 mouse embryonic stem cells (Robanus-Maandag *et al*, 1998) were cultured in mES knockout Dulbecco's modified Eagle's medium (Gibco) supplemented with 100 U/ml penicillin, 100 μg/ml streptomycin, 2 mM GlutaMAX, 1 mM sodium pyruvate, 1× non-essential amino acids, 100 μM β-mercaptoethanol (all from Gibco), 10% fetal calf serum and leukaemia inhibitory factor. Mouse ES cells were maintained by culturing them on

gelatin-coated plates containing irradiated primary mouse embryonic fibroblast feeder cells at 37°C and 5% $CO_2$. For clonogenic survival assays, mouse ES cells were cultured on gelatin-coated plates in Buffalo rat liver (BRL)-conditioned mES cell medium.

## Plasmids

Plasmids SpCas9(BB)-2A-GFP (PX458), U6-Chimeric_BB-CBh-SpCas9n-D10A (PX335) and spCas9-N863A (PX856) were a gift from Feng Zhang (Addgene plasmid #48138 and #42335, #62888, respectively). SpCas9-D10A-2A-GFP was generated via ligation of an *ApaI/AgeI*-digested fragment of plasmid PX335 into *ApaI/AgeI*-digested PX458. SpCas9-N863A-2A-GFP was generated via ligation of an *EcoRV/BsmI*-digested fragment of plasmid PX856 into *EcoRV/BsmI*-digested plasmid PX458. SpCas9-D10A-N863A-2A-GFP (nuclease-dead Cas9) was generated via ligation of an *EcoRV/BsmI*-digested fragment of plasmid Sp-Cas9-N863A-2A-GFP into *EcoRV/BsmI*-digested plasmid SpCas9-D10A-2A-GFP. To clone a target sequence into the PX backbones, two complementary oligonucleotides (Integrated DNA Technologies) with *BbsI* overhangs were phosphorylated, annealed and cloned in *BbsI* digested PX vectors as previously described (Cong *et al*, 2013). An overview of the targeted sequences can be found in Appendix Table S1.

## Transfections

Cells were transfected using Lipofectamine 2000 (Invitrogen) in a Lipofectamine:DNA ratio of 2.4:1 using an optimized protocol. Briefly, cells were trypsinized, counted and resuspended in mES knockout Dulbecco's modified Eagle's medium and $2 \times 10^6$ cells were transfected in suspension for 30 min at 37°C and 5% $CO_2$ in round-bottom tubes with 6 μg of total DNA, and subsequently, cells were seeded on gelatin-coated plates containing MEFs.

## Generation of knockout cell lines

*Polq*, *Ku80 Lig4*, *Poll* and *Polm* single knockout cell lines were generated by transfecting IB10 wild-type cells with plasmids co-expressing Cas9-WT-2A-GFP and sgRNAs. Cells were seeded at low density and maintained with regular medium changes for 8–10 days until colonies formed. Colonies were picked and grown in 96-well format, and at (semi)-confluence, cells were split to two 96-well plates. One plate was used for subculturing of cells; the other plate was used for DNA isolation and PCR analysis of the individual clones. Restriction fragment length polymorphism (RFLP, based on the loss of a unique restriction site) of PCR products was used to identify clones with a bi-allelic mutation, as previously described (Wang *et al*, 2013). The introduced bi-allelic mutations were recently described elsewhere (Zelensky *et al*, 2017). *Polq-Ku80* double-knockout cell lines were generated by targeting the *Polq* gene in Ku80-deficient cell lines. *Poll-Polm* double and *Polq-Poll-Polm* triple knockout cell lines were generated by targeting the *Poll* and *Polm* genes simultaneously in wild-type and *Polq*$^{-/-}$ cells, respectively. Experiments described in this study have been done using two independent clonally derived knockout lines per gene. All independently created and validated knockout clones of identical genotype behaved identical in all experiments tested. For purpose of reliability (increased sample size) and clarity of presenting, we combined the data of identical genotypes. For an overview of the targeted sequences and the oligonucleotides and restriction sites used for the RFLP analysis, see Appendix Table S1.

## Immunoblotting

Protein samples were separated on Novex 4–12% Bis-Tris gradient gels (Invitrogen) using MOPS SDS running buffer or on Novex 3–8% Tris-acetate gels (Invitrogen) using Tris-acetate SDS running buffer and transferred onto Immobilon-FL membranes (Merck Millipore). The primary antibodies used to analyse protein expression were as follows: anti-Ku80 (M-20 goat polyclonal, Santa Cruz sc-1485), anti-Lig4 (rabbit polyclonal, Abcam 80514), anti-Cas9 (mouse monoclonal, 7A9-3A3 Cell Signaling), anti-Tubulin (clone DM1a, Sigma), anti-polymerase mu (rabbit monoclonal, ab157465, Abcam) and anti-polymerase lambda (rabbit polyclonal, kind gift of Prof. Luis Blanco). The secondary antibodies CF680 goat anti-rabbit IgG and CF770 goat anti-mouse IgG (Biotium) and the Odyssey infrared imaging scanning system (LI-COR biosciences) were used to detect protein expression.

## Cell survival assay after ionizing radiation (IR)

Cells were trypsinized, counted and seeded at low density and exposed to IR using an YXlon X-ray generator (YXlon International) and left to grow for 7 days. Subsequently, cells were washed with 0.9% NaCl and stained with methylene blue to score the number of surviving colonies.

## *HPRT* gene mutation assay

Cells were transfected with the indicated Cas9-2A-GFP constructs and cultured for an additional 7 days (cells were passaged twice in this period). After 7 days, cells were trypsinized, counted and seeded at low density. For each sample, two plates were set up: one was left untreated to determine the cloning efficiency, whereas 5 μg/ml 6-thioguanine (6-TG) was added to the other plate to select for HPRT-deficient cells. Seven days after addition of 6-TG, plates were washed with 0.9% NaCl and stained with methylene blue. Surviving colonies were scored, and the *HPRT* mutation frequency was calculated as follows (Appendix Table S2):

$$\text{mutation frequency} = \frac{\text{number of 6-TG-resistant clones}}{\text{number of cells plated} \times \text{cloning efficiency}}$$

$$\left( \text{cloning efficiency} = \frac{\text{number of survived clones on untreated plates}}{\text{number of cells plated}} \right).$$

To obtain sequences of *HPRT* mutant clones, additional plates were set up with 6-TG selection. After 7–10 days of culturing, individual clones were picked and cultured in 96-well plates to (semi)-confluency. These plates were used for DNA isolation, and nested-PCRs were performed to amplify the targeted region; for an overview of the oligonucleotides used, see Appendix Table S1. Sequences were obtained by Sanger-sequencing PCR products using the forward oligonucleotide of the nested PCR.

**Cas9 survival assay**

Cells were transfected with the Cas9-2A-GFP encoding constructs, and cells were sorted 16 h after transfection for GFP expression via Fluorescence-activated cell sorting. GFP-positive cells were seeded at low density. After 7 days, plates were washed with 0.9% NaCl and stained with methylene blue. The survival of cells transfected with nuclease-dead Cas9 was calculated using the cloning efficiency of non-transfected corresponding cell lines and was set to 1.0 for wild-type cells.

**Bio-informatic analysis**

A custom Sanger Sequence-analyser was written (available upon request) to determine sequence alterations in the *HPRT* locus. Each Sanger sequence was filtered prior to comparison with a reference sequence on the following criteria: a stretch of $\geq$ 40nt was present where each base had an error probability of < 0.05 surrounding the sgRNA target site. All other nucleotides were masked. The filtered high-quality sequence was then compared to the reference sequence. Mutations were categorized into: SNV, deletion, insertion, deletion accompanied by an insertion (delins) or tandem duplication. For insertions $\geq$ 5nt, a longest common substring search was performed comparing the inserts to DNA sequences in the immediate vicinity of the sgRNA target (100 base pairs in both directions and orientation) to identify potential templates of inserted sequences. Additionally, a BLAST search was performed using the mouse genome or the sequence of plasmid PX458 to find the origin of insertions that could not be reliably mapped to the vicinity of the event. Insertions that completely mapped immediately adjacent to the junction of the event are annotated as tandem duplication.

**Expanded View** for this article is available online.

## Acknowledgements

We thank Dr. Sjef Verbeek for experimental advice on CRISPR/Cas9 and mouse embryonic stem cells. M.T. is supported by grants from the European Commission (DDResponse) and ZonMW/NGI-horizon. J.S. received support from the Dutch STW Open Technology Program and is funded by a VENI grant from NWO-Life Sciences.

## Author contributions

JS and MT designed the experiments. JS performed the majority of the experiments. HK performed experiments. RvS wrote the custom Sanger Sequence-analyzer and assisted in analysing the sequence data. JS analysed the data; JS and MT interpreted the results and wrote the manuscript and RvS commented on the manuscript.

## Conflict of interest

The authors declare that they have no conflict of interest.

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
