## [Review Process File · The EMBO Journal]

Manuscript EMBO-2017-96948

Mutational signatures of non-homologous and polymerase theta-mediated end-joining in embryonic stem cells

Joost Schimmel, Hanneke Kool, Mr. Robin van Schendel & Marcel Tijsterman

Corresponding author: Marcel Tijsterman, Leiden University Medical Center

Review timeline:

Submission date:	16 March 2017
Additional correspondence (Editor)	24 April 2017
Additional correspondence (Author)	25 April 2017
Editorial Decision:	27 April 2017
Revision received:	05 September 2017
Accepted:	29 October 2017

Editor: Hartmut Vodermaier

Transaction Report:

Additional Correspondence (Editor)

24 April 2017

Thank you again for submitting your manuscript EMBOJ-2017-96948, "Mutational signatures of non-homologous and polymerase Theta-mediated end joining in embryonic stem cells". We have now received comments from three expert referees, which I am enclosing copied below. As you will see, the reviewers express some overall interest, but also point out that there is considerable conceptual precedent from other recent studies, which are in addition not adequately acknowledged and discussed. Nevertheless, the referees indicate that the main novel finding (that Ku and Lig4 are important for mutagenic repair of breaks with long 5' or 3' protruding ssDNA ends) still represents an advance that may in principle warrant publication in The EMBO Journal, in case that it could be followed up in somewhat more detail. In this light, I would like to give you an opportunity to carefully consider the referees' comments and get back to us with a tentative point-by-point response letter, detailing how you could address their concerns in the eventual case of a revision. We would then take these responses into account for making our final decision on this manuscript. Given the general novelty concerns, it would be particularly helpful to hear whether and how you might be able to develop the key new aspect of tandem duplication generation by c-NHEJ for both 3' and 5' overhangs.

REFeree REPORTS

Referee #1:

In this study, the authors used CRISPR/Cas9 to generate a single DSB with near blunt ends in the HPRT locus to determine mutagenic outcomes in wild type, Polq^{-/-}, Ku80^{-/-}, Lig4^{-/-} and Polq^{-/-}

Ku80^{-/-}-mouse embryonic stem cell lines. The data presented provide strong support for Pol theta mediated repair (TMEJ) contributing to repair efficiency and repair outcome, even in cells proficient for c-NHEJ, and for redundancy between c-NHEJ and TMEJ for cell proliferation and repair. Most of the deletions recovered from c-NHEJ deficient cells had microhomology (MH) at the junctions, consistent with other studies, while end join events from Ku80^{-/-} Polq^{-/-} cells mostly lacked MH at the junctions, in agreement with MH dependence for TMEJ repair. Nuclease-deficient variants of Cas9 were used to determine whether alterations in DNA end structures (46 nucleotide 3' or 5' overhangs) change the genetic requirements for mutagenic repair. In the case of 5' protruding ends, most repair products had deletions removing part of the overhang, some of which were associated with tandem duplication of sequence corresponding to the overhang. Surprisingly, repair signatures from wild type and Polq^{-/-} cells were similar, suggesting NHEJ is largely responsible for these events. Most of the repaired products from the 3' overhang substrate retained most of the overhangs as tandem duplications; again, the spectra from wild type and Polq^{-/-} cells were similar. Thus for ends with very long protrusions, c-NHEJ appears to dominate repair products and TMEJ contributes redundantly to ensure repair and cell proliferation.

The studies presented support the conclusion that TMEJ acts in parallel, as well as redundantly, with c-NHEJ to repair DSBs in ES cells, and that TMEJ is mainly responsible for "alt-NHEJ". The recovery of c-NHEJ dependent tandem duplications from the substrates with long 3' or 5' protrusions is particularly interesting and provides new insight into the origin of this common class of mutagenic events. A couple of points the authors might want to discuss are: (1) the difficulty of assessing the contribution c-NHEJ to repair of breaks made by expression of nucleases in cells because of the problem with cycles of precise repair and re-cutting before mutating the cut site; (2) presumably, the ends produced by offset nicks may be held together for some time and could be acted on differently to extra-chromosomal linear DNAs transfected into cells (Wyatt et al, 2016).

Referee #2:

This is a review of EMBOJ-2017-96948, "Mutational signatures of non-homologous and polymerase Theta-mediated end joining in embryonic stem cells." In this report, mouse embryonic stem cells (ES) with mutations in the c-NHEJ factors Lig4 and Ku80, polymerase theta (Poltheta), and a Ku80/Poltheta double mutant, are examined for repair of CAS9-induced chromosomal breaks. The assay is to induce DNA breaks in different configurations at the Hprt locus, select for Hprt-clones, and then examine the products. There are two parts to the study: examining CAS9 DSBs induced by a single sgRNA, and with two sgRNAs creating various ssDNA overhangs. The more novel part of the study are the substrates with overhangs / ssDNA protrusions, in which the tandem duplications formed by such CAS9 experiments are shown to be dependent on Ku and Lig4, but not Poltheta, which if developed would be an important contribution for the field (major point 3). In contrast, I found that majority of this manuscript to be relatively incremental and with inadequate scholarship.

Major concerns:

1. A major limitation of this study is that a similar study with a single sgRNA was performed by Wyatt et. al. (PMID: 27453047) last year (see esp. Fig 4), which came to similar conclusions with Ku-Poltheta- double mutant cells. The findings of this publication are not comprehensively discussed in this current manuscript (except for a few mentions of the oligo substrates in this paper), which represents inadequate scholarship.
2. The methods are largely sound, although it is unclear how the authors can distinguish loss of mutagenic end joining vs. loss of clonogenic survival for the Ku-Poltheta double mutant, which is a key aspect of the study.
3. The findings that Ku and Lig4 are important for the tandem duplications caused by CAS9 staggered ends is certainly novel, but this part of the study is relatively limited. Defining how other polymerases and/or homologous recombination factors influence such TDs would improve the scope of the study. On a more positive note, I found the discussion of the prior literature demonstrating that CAS9-induced ssDNA protrusions causes tandem duplications to be comprehensive.

Minor concerns:

1. The authors should consider distinguishing between precise end joining and mutagenic end joining in their conclusion sentences, since without this distinction, the manuscript is confusing at points. For example, the abstract states that "TMEJ is most prevalent for DSBs that are near-blunt," but the result is that TMEJ is important for mutagenic repair of DSBs that are near-blunt.
2. The term "near blunt" should be described in detail. While CAS9 protein can cause 5' 1nt overhangs in biochemical experiments, products of end joining between two DSBs after CAS9 expression in mammalian cells are consistent with a high level of blunt DSBs (e.g. PMID: 26762978).
3. It is unclear why it was surprising to the authors that c-NHEJ deficient cells are not required for mutagenic end joining - this has been well established in the literature.
4. Stating that Boulton and Jackson 1996 was the first to define Alt-EJ may be controversial, given Roth and Wilson 1986 (PMID: 3025650).

Referee #3:

In this paper, Schimmel and colleagues have investigated the genetic requirements for mutagenesis following double- and single-strand break induction by CRISPR-Cas9 cleavage in mouse ES cells. They employ selection to quantify mutations in an HPRT reporter gene following breakage at two different locations. The relative usage of C-NHEJ and Pol theta-mediated end joining (TMEJ) for end-joining repair of different types of breaks has been recently reported by Wyatt et al. (2016) and mutagenic repair following CRISPR-Cas9 cleavage in human cancer cells has also been investigated (Bothmer et al., 2017). What sets the current investigation apart from those two studies is the use of mouse embryonic stem cells, which the authors hypothesized might behave differently from the mouse MEFs and human U2OS cell types utilized by the other groups. Indeed, while the previously reported function of Pol theta in end-joining repair is largely confirmed, the current study presents several surprising results, the most unexpected being that mutagenic repair of DSBs with 5' or 3' protruding ends occurs preferentially through C-NHEJ. The authors present models suggesting how C-NHEJ may be responsible for tandem duplications that are observed in these sequence contexts.

Overall, this paper makes an important contribution to the field by investigating the role of different DSB repair pathways to mutagenesis in ES cells, which could be important for CRISPR-based genetic therapies. However, I do think that the authors need to do a better job interpreting their results in the context of recent publications, particularly the two cited above. Explicit comparisons between the three studies should be made in the discussion, highlighting similarities and differences. Furthermore, the models for the generation of tandem duplications need to be better described (the model for the 5' overhang tandem duplications at the end of the discussion is helpful, but the nature and etiology of the 3' overhang duplications isn't clear to me). Finally, there are a large number of typos throughout the manuscript that need to be fixed.

Major issues:

1. For the experiments in Figures 1 and 3, the actual HPRT mutation frequencies need to be reported, in addition to the relative frequencies. These data could be included in a supplemental table.
2. One of the most surprising results in this paper was that DSBs with 3' protruding ends are preferentially repaired by C-NHEJ rather than TMEJ (Figure 3B). This is unexpected and runs counter to what was shown in Wyatt et al. and many other papers studying the role of Pol theta in alternative end joining using 3' overhangs. To explain this discrepancy, the authors suggest that there may be a difference between 3' ends generated during S phase from those generated by ionizing radiation-induced nicks. Why is ionizing radiation invoked here? I don't understand the logic-a more clear explanation is needed.

3. According to Figure S1A, it appears that the Polq frameshift mutations should introduce early stop codons and result in null alleles that abolish both ATPase and polymerase activity. However, this should really be confirmed with a Western blot, as was done for the other mutations.

4. Which of the mutants shown in Figure S1A are used for the assays throughout the rest of the paper should be clearly stated.

5. Pages 10, 11, and 14: It wasn't clear to me how the tandem deletions generated from the 5' overhangs differed from those observed with the 3' overhang substrate. Figure S5A shows a model for the 5' overhang substrate—a similar one is needed for 3' overhangs. In general, the section describing the tandem deletion products needs to be rewritten for clarity, as it represents a potential new type of repair with Cas9-induced breaks that is central to the novelty of this study.

Other points:

6. Figure 1G: what is the cellular survival of the lig4 mutants?

7. Throughout the paper, the authors refer to the Cas9-induced DSBs as "near-blunt." Is there a reason to assume that the DSBs do not have true blunt-ends? The sentence at the top of page 8 makes it sound like there could be small overhangs present after Cas9 cleavage.

8. Bottom of page 9: why would SSA not result in a mutagenic outcome? The logic isn't clear here.

9. For Figures S4 and S6, the overlays representing the insertions/duplications are nearly impossible to discern. Is there a better way to represent these?

Additional Correspondence (Author)

25 April 2017

Thanks again for handling our paper and for giving us the opportunity to respond to the comments of the reviewers. You will find a tentative point-to-point list below.

First of all, we were happy to read the generally positive and supportive comments. Perhaps it is worth mentioning that in addition to the recognized novelty concerning our tandem duplications (TDs) data our study also i) shows that, in embryonic cells, TMEJ acts on DSBs in cNHEJ proficient conditions - which provides an explanation for the alt-EJ signature of genomic scars in congenital disease, and ii) demonstrates that TMEJ represent almost all alt-EJ, a conclusion that follows from our double mutant analyses. In a new version we will rephrase the text of our manuscript and also discuss the mentioned Wyatt et al. and Bothmer et al., 2017 studies in more detail (as requested by refs 2 and 3) to point out those aspects better.

On the main novelty: you made clear that the cNHEJ involvement in repairing breaks with long protruding ssDNA ends, leading to TDs, represents the advance that is needed to warrant publication in EMBO J.

We will change the manuscript in two ways:

1) We will make this part of our study more prominent by i) putting more emphasis in abstract, introduction, results and discussion sections, ii) discussing the data and the inferred conclusion in more detail, iii) include a model figure to visually represent our data and propose the aetiology of DSB-induced TDs.

2) We will include new data, in which we address the potential involvement of the cNHEJ polymerases Mu and Lambda. We have successfully (confirmed by Westerns) generated knockout alleles of Lambda and Mu separately as well as made double mutant cells. We have analysed these for TD formation (as well as IR sensitivity). Our preliminary data, which needs further substantiation, indicates that both polymerases are involved in TD formation yet to a different extent. We also just generated Pol Mu Pol Lambda and Pol Theta triple mutant ES cells, thus lacking TMEJ and lacking the polymerases that are involved in cNHEJ. We will also include the analysis of those cells in a new version.

With these additions we believe that we will highlight the novel aspects of our work better but also bring the current analysis one step further. We thus hope that you will consider these changes sufficient to positively evaluate our manuscript.

There is however one question that I want to pose upfront, in order to optimally determine publication strategy of all our work: although our preliminary data argue for a role of both Mu and Lambda in cNHEJ TD formation we do not yet know the extend of their involvement - could be major but could also be more modest (we will know better after having analysed the triple mutant cells). This creates uncertainty as to the opinion of reviewer 2 on a resubmitted manuscript. Given the time investment, I want to ask you whether you can give me your opinion about this upfront?

1st Editorial Decision

27 April 2017

Thank you for response letter and proposal for improving your manuscript in response to the comments of our three referees. I have now carefully considered your responses and overall agree to your revision plans, which in my view should indeed be able to address all key concerns. With regard to the follow-up investigation of cellular Pol lambda and Pol mu knockouts, I appreciate that you are not able to anticipate the exact outcome of these experiments, but also feel that the inclusion of such experiments (as requested by referee 2) together with scholarly interpretation and discussions of their combined implications should go a long way to making the paper a more impactful candidate for The EMBO Journal (as even a lesser contribution of the additionally analyzed polymerases should probably allow you to better define what mechanisms may/may not be involved in the tandem duplication mechanism). Therefore, I would like to invite you to prepare and resubmit a manuscript revised along the discussed lines; should the proposed experiments require more time than our regular three-months revision period, I would be happy to offer an extension of this deadline in this case.

Thank you again for the opportunity to consider this work! I look forward to your revision.

1st Revision - authors' response

05 September 2017

Thanks again for giving us the opportunity to resubmit a revised version of this manuscript (see also our email correspondence at 27-04-2017) and also for extending the deadline for resubmission, which allowed us to include a substantial body of new experimental work.

In the new version, we addressed all comments and suggestions raised by the three reviewers which helped us to better our manuscript. A detailed point-to-point list is uploaded separately. The most important changes are:

- 1) We amended the main text to highlight the novelty of our finding that cNHEJ of DSBs with protruding ends can explain the formation of tandem duplications (TDs), which are frequently observed in mammalian genomes. We now also include a model figure (Fig 6H) to visually represent our data.
- 2) We extended our discussion section to discuss the recent work of Wyatt et al. 2016 and Bothmer et al., 2017.
- 3) We have included new work, in which we have generated knockout alleles of cNHEJ polymerases Lambda and Mu as well as made double and triple mutant (with pol theta) mES cells to address their potential involvement in TD formation. We found that these polymerases are not essential but play a subtler role: removal of these polymerases does not affect the overall effectivity of cNHEJ but does influence the mutagenic outcome: the mutation profiles in cells defective for Lambda and Mu were different from those in wild type cells. Our genetic data argues for a role of the cNHEJ polymerases specifically in the repair of DSBs with unaligned protruding tails (thus without having microhomology), and also point to a yet-to-discover polymerase activity within the cNHEJ pathway that acts on microhomology-aligned protrusions.

Response to the referees, point to point

We would like to thank all reviewers for their time and support, and for the constructive remarks and suggestions, which we feel helped us to increase the quality and readability of our manuscript.

Referee #1

(Report for Author)

In this study, the authors used CRISPR/Cas9 to generate a single DSB with near blunt ends in the HPRT locus to determine mutagenic outcomes in wild type, Polq^{-/-}, Ku80^{-/-}, Lig4^{-/-} and Polq^{-/-} Ku80^{-/-} mouse embryonic stem cell lines. The data presented provide strong support for Pol theta mediated repair (TMEJ) contributing to repair efficiency and repair outcome, even in cells proficient for c-NHEJ, and for redundancy between c-NHEJ and TMEJ for cell proliferation and repair. Most of the deletions recovered from c-NHEJ deficient cells had microhomology (MH) at the junctions, consistent with other studies, while end join events from Ku80^{-/-} Polq^{-/-} cells mostly lacked MH at the junctions, in agreement with MH dependence for TMEJ repair. Nuclease-deficient variants of Cas9 were used to determine whether alterations in DNA end structures (46 nucleotide 3' or 5' overhangs) change the genetic requirements for mutagenic repair. In the case of 5' protruding ends, most repair products had deletions removing part of the overhang, some of which were associated with tandem duplication of sequence corresponding to the overhang. Surprisingly, repair signatures from wild type and Polq^{-/-} cells were similar, suggesting NHEJ is largely responsible for these events. Most of the repaired products from the 3' overhang substrate retained most of the overhangs as tandem duplications; again, the spectra from wild type and Polq^{-/-} cells were similar. Thus for ends with very long protrusions, c-NHEJ appears to dominate repair products and TMEJ contributes redundantly to ensure repair and cell proliferation.

The studies presented support the conclusion that TMEJ acts in parallel, as well as redundantly, with c-NHEJ to repair DSBs in ES cells, and that TMEJ is mainly responsible for "alt-NHEJ". The recovery of c-NHEJ dependent tandem duplications from the substrates with long 3' or 5' protrusions is particularly interesting and provides new insight into the origin of this common class of mutagenic events. A couple of points the authors might want to discuss are: (1) the difficulty of assessing the contribution c-NHEJ to repair of breaks made by expression of nucleases in cells because of the problem with cycles of precise repair and re-cutting before mutating the cut site; (2) presumably, the ends produced by offset nicks may be held together for some time and could be acted on differently to extra-chromosomal linear DNAs transfected into cells (Wyatt et al, 2016).

The above-mentioned points are now commented upon in the discussion section of the manuscript

Referee #2

(Report for Author)

This is a review of EMBOJ-2017-96948, "Mutational signatures of non-homologous and polymerase Theta-mediated end joining in embryonic stem cells." In this report, mouse embryonic stem cells (ES) with mutations in the c-NHEJ factors Lig4 and Ku80, polymerase theta (Poltheta), and a Ku80/Poltheta double mutant, are examined for repair of CAS9-induced chromosomal breaks. The assay is to induce DNA breaks in different configurations at the Hprt locus, select for Hprt-clones, and then examine the products. There are two parts to the study: examining CAS9 DSBs induced by a single sgRNA, and with two sgRNAs creating various ssDNA overhangs. The more novel part of the study are the substrates with overhangs / ssDNA protrusions, in which the tandem duplications formed by such CAS9 experiments are shown to be dependent on Ku and Lig4, but not Poltheta, which if developed would be an important contribution for the field (major point 3). In contrast, I found that majority of this manuscript to be relatively incremental and with inadequate scholarship.

Major concerns:

1. A major limitation of this study is that a similar study with a single sgRNA was performed by Wyatt et. al. (PMID: 27453047) last year (see esp. Fig 4), which came to similar conclusions with

Ku-Poltheta- double mutant cells. The findings of this publication are not comprehensively discussed in this current manuscript (except for a few mentions of the oligo substrates in this paper), which represents inadequate scholarship.

We have extended the discussion section to repair this flaw.

2. The methods are largely sound, although it is unclear how the authors can distinguish loss of mutagenic end joining vs. loss of clonogenic survival for the Ku-Poltheta double mutant, which is a key aspect of the study.

We agree that we cannot discriminate between the two outcomes. Our data leads us to conclude that if cells lose the ability to repair a break through NHEJ or TMEJ they are (in fact) unable to produce a colony. Knocking out these 2 pathways thus strips cells of the ability to repair most Cas9-induced chromosomal breaks: loss of clonogenic survival is therefore compatible with the conclusion that EJ is essential for mutagenic repair. With the risk of stating the obvious we note that error free HR can only act on breaks induced in late S/G2 (provided that not both sisters are cut at the same time), however, through error-free repair the substrate for re-cutting is generated over and over.

3. The findings that Ku and Lig4 are important for the tandem duplications caused by CAS9 staggered ends is certainly novel, but this part of the study is relatively limited. Defining how other polymerases and/or homologous recombination factors influence such TDs would improve the scope of the study. On a more positive note, I found the discussion of the prior literature demonstrating that CAS9-induced ssDNA protrusions causes tandem duplications to be comprehensive.

We have rewritten parts on the paper to put more emphasis on the novel aspect of TD etiology and also included a model figure. More importantly, we assayed whether the cNHEJ polymerase Pol Lambda and Pol Mu were involved by generating and assaying poll^{-/-}, polm^{-/-} single, poll^{-/-} polm^{-/-} double, and poll^{-/-} polm^{-/-} polq^{-/-} triple mutant mES cells. These data are now presented (figure 6) and discussed in the manuscript.

Minor concerns:

1. The authors should consider distinguishing between precise end joining and mutagenic end joining in their conclusion sentences, since without this distinction, the manuscript is confusing at points. For example, the abstract states that "TMEJ is most prevalent for DSBs that are near-blunt," but the result is that TMEJ is important for mutagenic repair of DSBs that are near-blunt.

We have amended the text where appropriate.

2. The term "near blunt" should be described in detail. While CAS9 protein can cause 5' 1nt overhangs in biochemical experiments, products of end joining between two DSBs after CAS9 expression in mammalian cells are consistent with a high level of blunt DSBs (e.g. PMID: 26762978).

We have replaced "near-blunt" with "blunt" throughout the text. As the reviewer rightly points out blunt DSBs form the vast majority of break configuration induced by Cas9-wt. The mentioned reference has now been included.

3. It is unclear why it was surprising to the authors that c-NHEJ deficient cells are not required for mutagenic end joining - this has been well established in the literature.

We agree with the referee that published literature indeed points to mutagenic repair (alt-EJ) in the absence of cNHEJ, but the extend of this – that altEJ can be completely compensate for the loss of cNHEJ- was surprising to us. It is also a generally held conception that cNHEJ is responsible for mutagenic repair of CRISPR-induced breaks. We have now removed the subjective connotation "Surprisingly" as it is not needed.

4. Stating that Boulton and Jackson 1996 was the first to define Alt-EJ may be controversial, given Roth and Wilson 1986 (PMID: 3025650).

We thank the reviewer to point us towards this original work. We have now included a reference to it in the manuscript.

Referee #3

(Report for Author)

In this paper, Schimmel and colleagues have investigated the genetic requirements for mutagenesis following double- and single-strand break induction by CRISPR-Cas9 cleavage in mouse ES cells. They employ selection to quantify mutations in an HPRT reporter gene following breakage at two different locations. The relative usage of C-NHEJ and Pol theta-mediated end joining (TMEJ) for end-joining repair of different types of breaks has been recently reported by Wyatt et al. (2016) and mutagenic repair following CRISPR-Cas9 cleavage in human cancer cells has also been investigated (Bothmer et al., 2017). What sets the current investigation apart from those two studies is the use of mouse embryonic stem cells, which the authors hypothesized might behave differently from the mouse MEFs and human U2OS cell types utilized by the other groups. Indeed, while the previously reported function of Pol theta in end-joining repair is largely confirmed, the current study presents several surprising results, the most unexpected being that mutagenic repair of DSBs with 5' or 3' protruding ends occurs preferentially through C-NHEJ. The authors present models suggesting how C-NHEJ may be responsible for tandem duplications that are observed in these sequence contexts.

Overall, this paper makes an important contribution to the field by investigating the role of different DSB repair pathways to mutagenesis in ES cells, which could be important for CRISPR-based genetic therapies. However, I do think that the authors need to do a better job interpreting their results in the context of recent publications, particularly the two cited above. Explicit comparisons between the three studies should be made in the discussion, highlighting similarities and differences. Furthermore, the models for the generation of tandem duplications need to be better described (the model for the 5' overhang tandem duplications at the end of the discussion is helpful, but the nature and etiology of the 3' overhang duplications isn't clear to me).

We have now extended our discussion section to include comparison of our data to that of Wyatt et al., 2016 and Bothmer et al, 2017. We have now also put more emphasis to the generation of tandem duplication, for instance by including a model figure in the main body of the manuscript. To deepen our understanding about the molecular mechanism we have now include new data in which we analysed pol Mu and Pol Lambda knockouts, Pol Mu Pol Lambda double mutant cells and Pol Mu Pol Lambda Pol Theta triple mutant cells.

Finally, there are a large number of typos throughout the manuscript that need to be fixed.

We apologize for this. We have gone through the manuscript in great detail and also had it proofread by others to remove typos as much as possible.

Major issues:

1. For the experiments in Figures 1 and 3, the actual HPRT mutation frequencies need to be reported, in addition to the relative frequencies. These data could be included in a supplemental table.

This has been done (Supplemental table)

2. One of the most surprising results in this paper was that DSBs with 3' protruding ends are preferentially repaired by C-NHEJ rather than TMEJ (Figure 3B). This is unexpected and runs counter to what was shown in Wyatt et al. and many other papers studying the role of Pol theta in alternative end joining using 3' overhangs. To explain this discrepancy, the authors suggest that there may be a difference between 3' ends generated during S phase from those generated by ionizing radiation-induced nicks. Why is ionizing radiation invoked here? I don't understand the logic-a more clear explanation is needed.

The phrase leading to the confusion has been removed: being in the results section it was misplaced anyhow. We rephrased some of the sentences in the discussion section on this point trying to get our thoughts across clearly.

3. According to Figure S1A, it appears that the Polq frameshift mutations should introduce early stop codons and result in null alleles that abolish both ATPase and polymerase activity. However, this should really be confirmed with a Western blot, as was done for the other mutations.

Unfortunately, none of the available antibodies to human Pol theta detect mouse Pol Theta (our unpublished observations and also pointed out by other studies (e.g. Yousefzadeh et al., (2014) and Wyatt et al.,2016). We have here generated and used multiple alleles of Pol Theta, targeting different exons, all behaved identical in all assays analysed. We also have generated a mutation within the polymerase domain, which is as sensitive to IR as the early stop alleles (data not shown). And we have, in an earlier manuscript, validated these polq-/-cell lines by complementation experiments with wildtype cDNA; this is now also mentioned in the manuscript.

4. Which of the mutants shown in Figure S1A are used for the assays throughout the rest of the paper should be clearly stated.

All independently created and validated knockout clones of identical genotype behaved identical in all experiments tested. For purpose of reliability (to increase sample size) and clarity (of presenting) we combined the data of one genotype and show this as a single value. Experiments are nevertheless done multiple times including both independent lines resulting in n = 4 or n = 6 (see also Table S2). This has now been clearly indicated in the materials and method section.

5. Pages 10, 11, and 14: It wasn't clear to me how the tandem deletions generated from the 5' overhangs differed from those observed with the 3' overhang substrate. Figure S5A shows a model for the 5' overhang substrate-a similar one is needed for 3' overhangs. In general, the section describing the tandem deletion products needs to be rewritten for clarity, as it represents a potential new type of repair with Cas9-induced breaks that is central to the novelty of this study.

We have amended the text of the manuscript at various positions in the results and discussion sections to improve clarity. Moreover, we have now included a model figure in the main body of the manuscript.

Other points:

6. Figure 1G: what is the cellular survival of the lig4 mutants?

We now have included this data.

7. Throughout the paper, the authors refer to the Cas9-induced DSBs as "near-blunt." Is there a reason to assume that the DSBs do not have true blunt-ends? The sentence at the top of page 8 makes it sound like there could be small overhangs present after Cas9 cleavage.

We have removed the "near-"-connotation from the manuscript, as indeed the vast majority of DSBs induced by wildtype Cas9 are blunt.

8. Bottom of page 9: why would SSA not result in a mutagenic outcome? The logic isn't clear here.

We have rephrased the text to be clearer. The remark aims to point out that SSA of the substrates generated by Cas9 nickases will be error free: SSA normally results in the removal of one copy of the homologous sequence (that is the basis for the annealing) as well as any intervening sequence. In this case, there is no intervening sequence and the two homologous sequences are generated from a single sequence tract. Hence, SSA restores the original sequence.

9. For Figures S4 and S6, the overlays representing the insertions/duplications are nearly impossible to discern. Is there a better way to represent these?

We've changed the colour schemes to make the insertions more clearly visible.

2nd Editorial Decision

22 October 2017

Thank you for submitting your revised manuscript for our consideration.

It has now been seen once more by two of the original referees (see comments below), and I am pleased to inform you that both are largely satisfied and have no further objections towards publication. We shall therefore be happy to accept the study for The EMBO Journal, pending a few minor modifications to the text.

Please incorporate the remaining specific points raised by referee 3 into the text as appropriate. Please also consider the re-introduction of the previous figure mentioned in this report (in which case it would need also a callout in the text).

REFeree REPORTS

Referee #1

(Report for Author)

The authors have adequately addressed my comments on the original submission. I am fully supportive of publication.

Referee #3

(Report for Author)

Overall, the authors have done a nice job responding to the reviewer queries and criticisms. I feel that they have now interpreted their results more clearly in the context of the existing $\text{pol } \theta$ and CRISPR literature. The addition of the $\text{pol } \mu$ and $\text{pol } \lambda$ data doesn't provide much mechanistic information but is nonetheless a nice addition to the study. The Figure in 6H is also helpful. I have a few small items that should still be addressed, but overall I think this work provides some nice insight into the etiology of Cas9-induced DSB repair products in mouse embryonic stem cells.

1. It appears that the old Figure S1, showing the exact mutations in each of the repair genes, has been removed. This information was useful and should be included.
2. The 'near-blunt' terminology is still used in Figure EV1 and on page 14.
3. Page 10: "We suspect that the single case in $\text{Polq}^{-/-}$ cells that is annotated as templated insert to be the result of a deletion formed after a re-cutting of a TD outcome..." Do you mean $\text{lig4}^{-/-}$ cells?
4. First paragraph of discussion: "We demonstrate that TMEJ and cNHEJ together constitute the error-prone mechanisms by which embryonic stem cells repair DSBs, in their absence cells become extremely sensitive to IR-induced DSBs. It is important to note that cNHEJ could also be an error-free mechanism for repair of IR-induced DSBs and that this function could also explain the extreme sensitivity.
5. The final sentence in the discussion should also cite Wood and Doublet, 2016.
6. The manuscript still needs to be proofread carefully for English grammar errors.

Please find attached the final version of our manuscript, a response listing the final changes and one revised figure. Let me know if anything else needs to be done or anything is unclear.

Referee #3

(Report for Author)

Overall, the authors have done a nice job responding to the reviewer queries and criticisms. I feel that they have now interpreted their results more clearly in the context of the existing pol theta and CRISPR literature. The addition of the pol mu and pol lambda data doesn't provide much mechanistic information but is nonetheless a nice addition to the study. The Figure in 6H is also helpful. I have a few small items that should still be addressed, but overall I think this work provides some nice insight into the etiology of Cas9-induced DSB repair products in mouse embryonic stem cells.

1. It appears that the old Figure S1, showing the exact mutations in each of the repair genes, has been removed. This information was useful and should be included.

-> The supplemental Figure was removed because the details of the alleles have recently been published elsewhere (Zelensky et al., Nat comm 2017). We refer to that study and the alleles used in the Methods section.

2. The 'near-blunt' terminology is still used in Figure EV1 and on page 14.

-> This is now fixed; a new Figure EV1 is attached

3. Page 10: "We suspect that the single case in Polq^{-/-} cells that is annotated as templated insert to be the result of a deletion formed after a re-cutting of a TD outcome..." Do you mean lig4^{-/-} cells?

-> No, we here refer to the templated insert (in trans) displayed in fig 4D.

4. First paragraph of discussion: "We demonstrate that TMEJ and cNHEJ together constitute the error-prone mechanisms by which embryonic stem cells repair DSBs, in their absence cells become extremely sensitive to IR-induced DSBs.

It is important to note that cNHEJ could also be an error-free mechanism for repair of IR-induced DSBs and that this function could also explain the extreme sensitivity.

-> This has now been done

5. The final sentence in the discussion should also cite Wood and Doublet, 2016.

-> The reference has now been included

6. The manuscript still needs to be proofread carefully for English grammar errors.

-> This has been done to the best of our abilities.

Corresponding Author Name: Marcel Tijsterman

Journal Submitted to: The EMBO journal

Manuscript Number: EMBOJ-2017-96948